# Large, three-generation human families reveal post-zygotic mosaicism and variability in germline mutation accumulation

Thomas A Sasani[1]*, Brent S Pedersen[1], Ziyue Gao[2], Lisa Baird[1], Molly Przeworski[3,4], Lynn B Jorde[1,5]*, Aaron R Quinlan[1,5,6]*

[1]Department of Human Genetics, University of Utah, Salt Lake City, United States; [2]Howard Hughes Medical Institute and Department of Genetics, Stanford University, Stanford, United States; [3]Department of Biological Sciences, Columbia University, New York City, United States; [4]Department of Systems Biology, Columbia University, New York City, United States; [5]USTAR Center for Genetic Discovery, University of Utah, Salt Lake City, United States; [6]Department of Biomedical Informatics, University of Utah, Salt Lake City, United States

**Abstract** The number of de novo mutations (DNMs) found in an offspring's genome increases with both paternal and maternal age. But does the rate of mutation accumulation in human gametes differ across families? Using sequencing data from 33 large, three-generation CEPH families, we observed significant variability in parental age effects on DNM counts across families, ranging from 0.19 to 3.24 DNMs per year. Additionally, we found that ~3% of DNMs originated following primordial germ cell specification in a parent, and differed from non-mosaic germline DNMs in their mutational spectra. We also discovered that nearly 10% of candidate DNMs in the second generation were post-zygotic, and present in both somatic and germ cells; these gonosomal mutations occurred at equivalent frequencies on both parental haplotypes. Our results demonstrate that rates of germline mutation accumulation vary among families with similar ancestry, and confirm that post-zygotic mosaicism is a substantial source of human DNM.
DOI: https://doi.org/10.7554/eLife.46922.001

*For correspondence:
tom.sasani@utah.edu (TAS);
lbj@genetics.utah.edu (LBJ);
aquinlan@genetics.utah.edu (ARQ)

## Introduction

In a 1996 lecture at the National Academy of Sciences, James Crow noted that 'without mutation, evolution would be impossible' (*Crow, 1997*). His remark highlights the importance of understanding the rate at which germline mutations occur, the mechanisms that generate them, and the effects of gamete-of-origin and parental age. Not surprisingly, continued investigation into the germline mutation rate has helped to illuminate the timing and complexity of human evolution and demography, as well as the key role of spontaneous mutation in human disease (*Scally and Durbin, 2012*; *Moorjani et al., 2016*; *Deciphering Developmental Disorders Study, 2017*; *Yuen et al., 2016*; *Acuna-Hidalgo et al., 2016*; *Veltman and Brunner, 2012*).

Some of the first careful investigations of human mutation rates can be attributed to J.B.S. Haldane and others, who cleverly leveraged an understanding of mutation-selection balance to estimate rates of mutation at individual disease-associated loci (*Haldane, 1935*; *Nachman, 2008*). Over half of a century later, phylogenetic analyses inferred mutation rates from the observed sequence divergence between humans and related primate species at a small number of loci (*Nachman and Crowell, 2000*; *Shendure and Akey, 2015*). In the last decade, whole genome sequencing of pedigrees

**eLife digest** Humans receive half of their DNA from each of their parents. However, this inherited DNA is not identical to the corresponding half of the parents' genetic material. Instead, both the egg and the sperm that combine to generate an embryo carry so-called 'germline de novo' mutations that are not present in the rest of the parents' cells. Although these de novo mutations are an important source of genetic diversity, they can also cause disease.

Geneticists have a longstanding interest in how, when and at what rate germline de novo mutations arise. These questions are commonly addressed by analyzing the DNA of large cohorts of two-generation families. Now, Sasani et al. have used the genetic data of 33 families in Utah, United States, which all span three generations, to determine the rate at which de novo mutations appear.

The analysis revealed that, on average, each person has around 70 de novo mutations that were not present in their parent's genetic code. Sasani et al. also found that sperm and egg cells from older parents typically contain more de novo mutations. However, this effect varied substantially across the Utah families. In some families, an increase of one year in the parents' age resulted in over three extra de novo mutations in their children. In others, the number of new mutations barely increased at all.

In addition, Sasani et al. found that almost 10% of de novo mutations do not occur in the parents' sperm or eggs, but happen in the embryo very soon after fertilization. These mutations can lead to 'mosaicism', resulting in a person having a mutation in some, but not all of their organs and tissues. In some cases, this could cause an unknown number of sperm and egg cells to carry a mutation that others do not. This makes it hard to predict how likely two or more siblings are to inherit the mutation.

This analysis reveals that parental age affects the number of de novo mutations in children, but this effect changes from family to family. This finding could point to genetic or environmental factors that alter the human mutation rate.

DOI: https://doi.org/10.7554/eLife.46922.002

has enabled direct estimates of the human germline mutation rate by identifying mutations present in offspring yet absent from their parents (de novo mutations, DNMs) (*Ségurel et al., 2014*; *Scally and Durbin, 2012*; *Jónsson et al., 2017*; *Goldmann et al., 2016*; *Kong et al., 2012*; *Roach et al., 2010*; *Francioli et al., 2015*). Numerous studies have employed this approach to analyze the mutation rate in cohorts of small, nuclear families, producing estimates nearly two-fold lower than those from phylogenetic comparison (*Roach et al., 2010*; *Kong et al., 2012*; *Jónsson et al., 2017*; *Goldmann et al., 2016*; *Scally and Durbin, 2012*; *Shendure and Akey, 2015*; *Turner et al., 2017*).

These studies have demonstrated that the number of DNMs increases with both maternal and paternal ages; such age effects can likely be attributed to a number of factors, including the increased mitotic divisions in sperm cells following puberty, an accumulation of damage-associated mutation, and substantial epigenetic reprogramming undergone by germ cells (*Jónsson et al., 2017*; *Kong et al., 2012*; *Goldmann et al., 2016*; *Rahbari et al., 2016*; *Crow, 2000*; *Gao et al., 2019*). There is also evidence that the mutational spectra of de novo mutations differ in the male and female germlines (*Jónsson et al., 2017*; *Goldmann et al., 2016*; *Francioli et al., 2015*; *Gao et al., 2019*; *Agarwal and Przeworski, 2019*). Furthermore, a recent study of three two-generation pedigrees, each with 4 or five children, indicated that paternal age effects may differ across families (*Rahbari et al., 2016*). However, two-generation families with few offspring provide limited power to quantify parental age effects on mutation rates and restrict the ability to assign a gamete-of-origin to ~20–30% of DNMs (*Rahbari et al., 2016*; *Jónsson et al., 2017*; *Goldmann et al., 2016*).

Here, we investigate germline mutation among families with large numbers of offspring spanning many years of parental age. We describe de novo mutation dynamics across multiple births using blood-derived DNA samples from large, three-generation families from Utah, which were collected as part of the Centre d'Etude du Polymorphisme Humain (CEPH) consortium (*Dausset et al., 1990*). The CEPH/Utah families have played a central role in our understanding of human genetic variation (*Prescott et al., 2008*; *1000 Genomes Project Consortium et al., 2015*) by guiding the

construction of reference linkage maps for the Human Genome Project (*Lander et al., 2001*), defining haplotypes in the International HapMap Project (*International HapMap Consortium, 2003*), and characterizing genome-wide variation in the 1000 Genomes Project (*1000 Genomes Project Consortium et al., 2015*).

The CEPH/Utah pedigrees are uniquely powerful for the study of germline mutation dynamics in that they have considerably more (min = 4, max = 16, median = 8) offspring than those used in many prior studies of the human mutation rate (*Supplementary file 1*). Multiple offspring, whose birth dates span up to 27 years, motivated our investigation of parental age effects on DNM counts within families and allowed us to ask whether these effects differed across families. The structure of all CEPH/Utah pedigrees (*Supplementary file 1*) also enables the use of haplotype sharing through three generations to determine the parental haplotype of origin for nearly all DNMs in the second generation. Using this large dataset of 'phased' DNMs, we can investigate the effects of gamete-of-origin on human germline mutation in greater detail.

Finally, if a DNM occurs in the early cell divisions following zygote fertilization (considered gonosomal), or during the proliferation of primordial germ cells, it may be mosaic in the germline of that individual. This mosaicism can then present as recurrent DNMs in two or more children of that parent. As DNMs are an important source of genetic disease (*Campbell et al., 2014b*; *Campbell et al., 2015*; *Biesecker and Spinner, 2013*; *Forsberg et al., 2017*; *Acuna-Hidalgo et al., 2016*; *Veltman and Brunner, 2012*), it is critical to understand the rates of mosaic DNM transmission in families. The structures of the CEPH/Utah pedigrees enable the identification of these recurrent DNMs and can allow one to distinguish mutations arising as post-zygotic gonosomal variants from those that are mosaic in the germline of the second generation.

## Results

### Identifying high-confidence DNMs using transmission to a third generation

We sequenced the genomes of 603 individuals from 33, three-generation CEPH/Utah pedigrees to a genome-wide median depth of ~30X (*Figure 1—figure supplement 1*, *Supplementary file 1*), and removed 10 samples from further analysis following quality control using peddy (*Pedersen and Quinlan, 2017a*). After standard quality filtering, we identified a total of 4,671 germline de novo mutations in 70 second-generation individuals, each of which was transmitted to at least one offspring in the third generation (*Figure 1a*, *Supplementary file 2*). Approximately 92% (4,298 of 4,671) of DNMs observed in the second generation were single nucleotide variants (SNVs), and the remainder were small (<=10 bp) insertion/deletion variants. The eight parents of four second-generation samples were re-sequenced to a median depth of ~60X (*Figure 1—figure supplement 1d*), allowing us to estimate a false positive rate of 4.5% for our de novo mutation detection strategy (Materials and methods). Taking all second-generation samples together, we calculated median germline mutation rates of $1.10 \times 10^{-8}$ and $9.29 \times 10^{-10}$ per base pair per generation for SNVs and indels, respectively, which corroborate prior estimates based on family genome sequencing with roughly comparable parental ages (*Jónsson et al., 2017*; *Kong et al., 2012*; *Besenbacher et al., 2016*; *Rahbari et al., 2016*). Extrapolating to a diploid genome size of ~6.4 Gbp, we therefore estimate an average number of 70.1 de novo SNVs and 5.9 de novo indels per genome, at average paternal and maternal ages of 29.1 and 26.0 years, respectively (*Sasani, 2019*).

### Parent-of-origin and parental age effects on de novo mutation observed in the second generation

We determined the parental gamete-of-origin for a median of 98.5% of de novo variants per second-generation individual (range: 90.3–100%) by leveraging haplotype sharing across all three generations in a family (*Kong et al., 2012*; *Jónsson et al., 2017*), as well as read tracing of DNMs to informative sites in the parents (*Figure 1b*, *Figure 1—figure supplement 2*). The ratio of paternal to maternal DNMs was 3.96:1, and 79.8% of DNMs were paternal in origin. We then measured the relationship between the number of phased DNMs observed in each child and the ages of the child's parents at birth (*Figure 2a*). After fitting Poisson regressions, we observed a significant paternal age effect of 1.44 (95% CI: 1.12–1.77, p<2e-16) additional DNMs per year, and a significant maternal

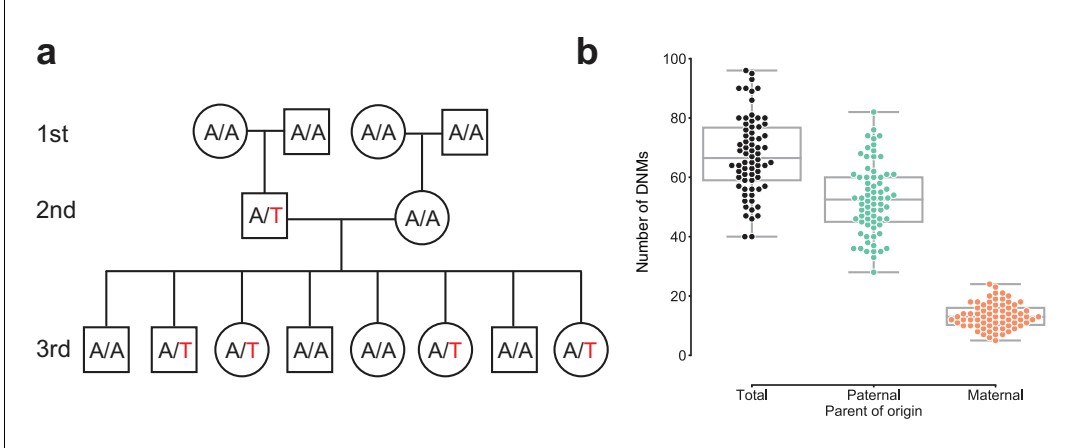

**Figure 1.** Estimating the rate of germline mutation using multigenerational CEPH/Utah pedigrees. (**a**) The CEPH/Utah dataset comprises 33 three-generation families. Summaries of sequencing coverage for CEPH/Utah individuals are presented in *Figure 1—figure supplement 1*. After identifying candidate de novo mutations in the second generation (e.g., the de novo 'T' mutation shown in the second-generation father), it is possible to assess their validity both by their absence in the parental (first) generation and by transmission to one or more offspring in the third generation. (**b**) Total numbers of DNMs (both SNVs and indels) identified across second-generation CEPH/Utah individuals and stratified by parental gamete-of-origin. Boxes indicate the interquartile range (IQR), and whiskers indicate 1.5 times the IQR. Diagrams of phasing strategies for germline DNMs are presented in *Figure 1—figure supplement 2*.

DOI: https://doi.org/10.7554/eLife.46922.003

The following figure supplements are available for figure 1:

**Figure supplement 1.** Distribution of sequencing coverage in CEPH/Utah samples (**a**) The fraction of bases greater than or equal to the specified coverage in the second generation, (**b**) third generation, (**c**) first-generation parents sequenced to 30X coverage, and (**d**) first-generation parents re-sequenced to 60X coverage.

DOI: https://doi.org/10.7554/eLife.46922.004

**Figure supplement 2.** Determining the parent-of-origin for de novo mutations using transmission.

DOI: https://doi.org/10.7554/eLife.46922.005

age effect of 0.38 (95% CI: 0.21–0.55, p=1.24e-5) DNMs per year (*Figure 2a*). These confirm prior estimates of the paternal and maternal age effects on de novo mutation accumulation, and further suggest that both older mothers and fathers contribute to increased DNM counts in children (*Figure 2—figure supplement 1*) (*Jónsson et al., 2017*; *Goldmann et al., 2016*; *Rahbari et al., 2016*; *Wong et al., 2016*; *Besenbacher et al., 2015*).

We next compared the paternal and maternal fractions of phased autosomal DNMs identified in the second generation across eight mutational classes (*Figure 2b*). In maternal mutations, there was an enrichment of C > T transitions in a non-CpG context (p=7.65e-6, Chi-squared test of independence), and we observed an enrichment of T > G transversions in paternal mutations (p=4.93e-3, Chi-squared test of independence). Maternal and paternal enrichments of C > T and T > G, respectively, have been reported in recent studies of de novo mutation spectra, though the mechanisms underlying these observations are currently unclear (*Goldmann et al., 2016*; *Jónsson et al., 2017*). We additionally stratified second-generation individuals by the ages of their parents at birth and found no significant differences in the mutational spectra of children born to older or younger parents, though we may be underpowered to detect these differences in our dataset (*Figure 2—figure supplement 2*).

## Evidence for inter-family variability of parental age effects on offspring DNM counts

A recent study of three two-generation pedigrees with multiple offspring suggested that the effect of paternal age on DNM counts in children may differ between families (*Rahbari et al., 2016*). Given the large numbers of offspring in the CEPH/Utah pedigrees, we were motivated to perform an investigation of parental age effects on mutation counts within individual families. To measure these effects in the CEPH dataset, we first generated a high-quality set of de novo variants observed in

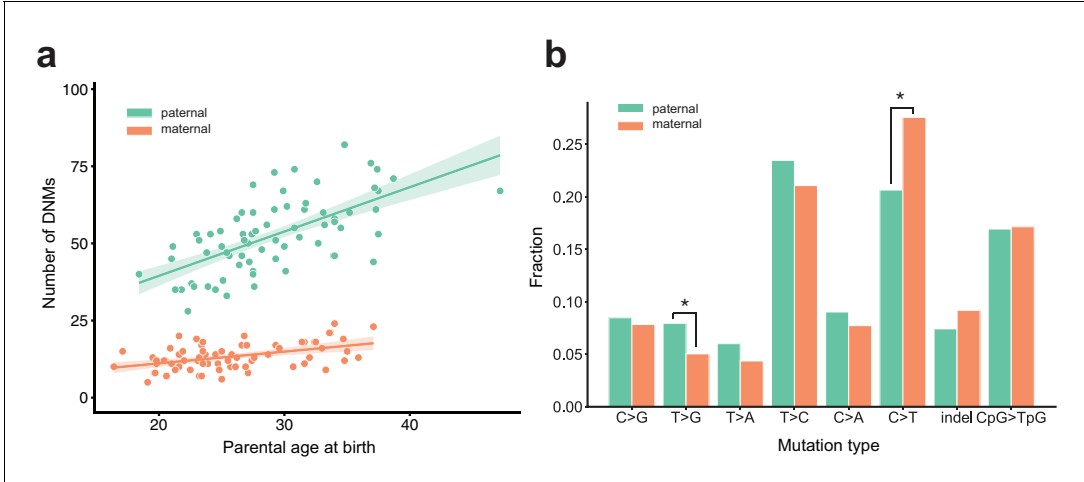

**Figure 2.** Effects of parental age and sex on autosomal DNM counts and mutation types in the second generation. (a) Numbers of phased paternal and maternal de novo variants as a function of parental age at birth. Poisson regressions (with 95% confidence bands, calculated as 1.96 times the standard error) were fit for mothers and fathers separately using an identity link. Germline mutation rates, as a function of both paternal and maternal ages, are presented in *Figure 2—figure supplement 1*. (b) Mutation spectra in autosomal DNMs phased to the paternal (n = 3,584) and maternal (n = 880) haplotypes. Asterisks indicate significant differences between paternal and maternal fractions at a false-discovery rate of 0.05 (Benjamini-Hochberg procedure), using a Chi-squared test of independence. P-values for each comparison are: C > G: 0.719, T > G: 4.93e-3, T > A: 8.60e-2, T > C: 8.02e-2, C > A: 0.159, C > T: 7.65e-6, indel: 8.01e-2, CpG >TpG: 0.835. Mutation spectra stratified by parental ages are presented in *Figure 2—figure supplement 2*.

DOI: https://doi.org/10.7554/eLife.46922.006

The following figure supplements are available for figure 2:

**Figure supplement 1.** Contribution of maternal and paternal age to de novo mutation rates.
DOI: https://doi.org/10.7554/eLife.46922.007

**Figure supplement 2.** Comparison of mutation spectra in children born to older or younger parents.
DOI: https://doi.org/10.7554/eLife.46922.008

the third generation, excluding recurrent (mosaic) DNMs shared by multiple third-generation siblings, likely post-zygotic DNMs (Materials and methods), and 'missed heterozygotes' in the second generation (0.4% of heterozygous variants). The 'missed heterozygotes' represent apparent DNMs in the third generation that were, in fact, likely inherited from a second-generation parent who was incorrectly genotyped as being homozygous for the reference allele (Materials and methods). In total, we detected 24,975 de novo SNVs and small indels in 350 individuals in the third generation (*Supplementary file 3*). Of these, we were able to confidently determine a parental gamete-of-origin for 5,336 (median of 21% per third-generation individual; range of 8–38%) using read tracing, and assign 4,201 (78.7%) of these to fathers. Given the comparatively low phasing rate in the third generation, we focused our age effect analysis on the relationship between paternal age only and the total number of autosomal DNMs in each individual, regardless of parent-of-origin. Taking all third-generation individuals into account, we estimate the slope of the paternal age effect to be 1.72 DNMs per year (95% CI: 1.58–1.85, p<2e-16). Within a given family, maternal and paternal ages are perfectly correlated; therefore, the paternal effect approximates the combined age effects of both parents.

When inspecting each family separately, we observed a wide range of paternal age effects among the CEPH/Utah families (*Figure 3*). To test whether these observed effects varied significantly between families, we fit a Poisson regression that incorporated the effects of paternal age, family membership, and an interaction between paternal age and family membership, across all third-generation individuals in CEPH/Utah pedigrees. As a small number of the CEPH/Utah pedigrees comprise multiple three-generation families (*Supplementary file 1*), we assigned each unique set of second-generation parents and their third-generation children a distinct ID, resulting in a total of 40 families (*Figure 3—figure supplement 1*). Overall, the effect of paternal age on offspring DNM counts varied widely across pedigrees, from only 0.19 (95% CI: −1.05–1.44) to nearly 3.24 (95% CI:

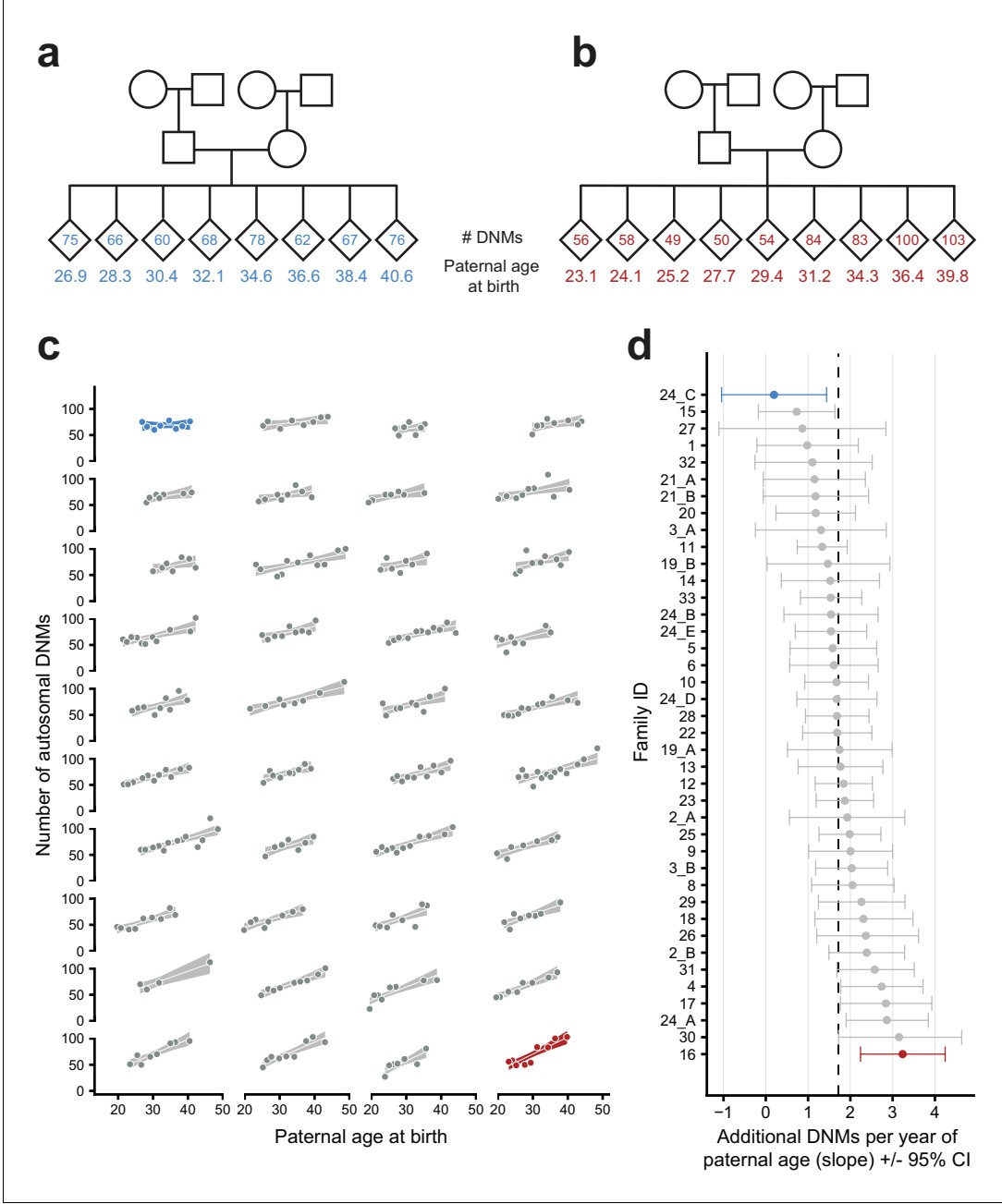

**Figure 3.** Parental age effects on autosomal germline mutation counts vary significantly among CEPH/Utah families. Illustrations of pedigrees exhibiting the smallest (family 24_C, panel **a**) and largest (family 16, panel **b**) paternal age effects on third-generation DNM counts demonstrate the extremes of inter-family variability. Diamonds are used to anonymize the sex of each third-generation individual. The method used to separate CEPH/Utah pedigrees into unique groups of second-generation parents and third-generation children is presented in *Figure 3—figure supplement 1*. Third-generation individuals are arranged by birth order from left to right. The number of autosomal DNMs observed in each third-generation individual is shown within the diamonds, and the age of the father at the third-generation individual's birth is shown below the diamond. The coloring for these two families is used to identify them in panels c and d. (**c**) The total number of autosomal DNMs is plotted versus paternal age at birth for third-generation individuals from all CEPH/Utah families. Regression lines and 95% confidence bands indicate the predicted number of DNMs as a function of paternal age using a Poisson regression (identity link). Families are sorted in order of increasing slope, and families with the least and greatest paternal age effects are highlighted in blue and red, respectively. (**d**) A Poisson regression (predicting autosomal DNMs as a function of paternal age) was fit to each family separately; the slope of each family's

*Figure 3 continued on next page*

*Figure 3 continued*

regression is plotted, as well as the 95% confidence interval of the regression coefficient estimate. The same two families are highlighted as in (a). A dashed black line indicates the overall paternal age effect (estimated using all third-generation samples). Families are ordered from top to bottom in order of increasing slope, as in (c). A random sampling approach was used to assess the robustness of the per-family regressions to possible outliers; the results of these simulations are shown in *Figure 3—figure supplement 2*.

DOI: https://doi.org/10.7554/eLife.46922.009

The following figure supplements are available for figure 3:

**Figure supplement 1.** Defining unique families in the CEPH/Utah dataset.

DOI: https://doi.org/10.7554/eLife.46922.010

**Figure supplement 2.** Paternal age effect ranks of CEPH/Utah families are robust to outlier samples.

DOI: https://doi.org/10.7554/eLife.46922.011

---

2.24–4.24) additional DNMs per year. A goodness-of-fit test supported the use of a 'family-aware' regression model when compared to a model that ignores family membership, even after accounting for variable sequencing coverage across third-generation samples (median autosomal base pairs covered = 2,582,875,060; ANOVA: p=9.36e-10). Moreover, we found that the interaction between paternal age and family membership improved the fit of the linear model (p=0.043, *Appendix 1— table 1*), suggesting that inter-family variability involves differences in paternal age effects (i.e., the slopes of each regression). We note that the confidence intervals surrounding the slope point estimates for some CEPH/Utah families are quite wide, likely due to the small number of third-generation individuals (with respect to count-based regression) in each family, as well as some stochastic noise in the DNM counts attributed to each child (*Figure 3d*). Nonetheless, family rankings based upon the effect of paternal age on DNM counts are stable and relatively insensitive to outliers (*Figure 3—figure supplement 2*).

Finally, when compared to a multiple regression that includes the effects of both paternal and maternal age, a model that takes family membership into account remained a significantly better fit (ANOVA: p=2.12e-5). The high degree of correlation between paternal and maternal ages makes it difficult to tease out the individual contributions of each parent to the observed inter-family differences. Nonetheless, these results suggest the existence of substantial variability in parental age effects across CEPH/Utah families, which could involve both genetic and environmental factors that differ among families.

## Identifying gonadal, post-primordial germ cell specification (PGCS) mosaicism in the second generation

Generally, studies of de novo mutation focus on variants that arise in a single parental gamete. However, if a de novo variant arises during or after primordial germ cell specification (PGCS), that variant may be present in multiple resulting gametes and absent from somatic cells (*Rahbari et al., 2016*; *Acuna-Hidalgo et al., 2015*; *Campbell et al., 2014b*; *Tang et al., 2016*; *Jónsson et al., 2018*; *Campbell et al., 2015*; *Biesecker and Spinner, 2013*). These variants can therefore be present in more than one offspring as apparent de novo mutations. In each family, we searched for post-PGCS germline mosaic variants by identifying high-confidence DNMs that were shared by two or more third-generation individuals, and were absent from the blood DNA of any parents or grandparents within the family (*Figure 4a*). Given the large number of third-generation siblings in each CEPH/Utah family, we had substantially higher power to detect germline mosaicism that occurred in in the second generation than in prior studies. In total, we identified 720 single-nucleotide germline mosaic mutations at a total of 303 unique sites, which were subsequently corroborated through visual inspection using the Integrative Genomics Viewer (IGV) (*Supplementary file 4*) (*Thorvaldsdóttir et al., 2013*). Of the phased shared germline mosaic mutations, 124/260 (47.7%) were paternal in origin; thus, the mutations that occurred following PGCS likely occurred irrespective of any parental sex biases on mutation counts. Overall, approximately 3.1% (720/23,399) of all single-nucleotide DNMs observed in the third generation likely arose during or following PGCS in a

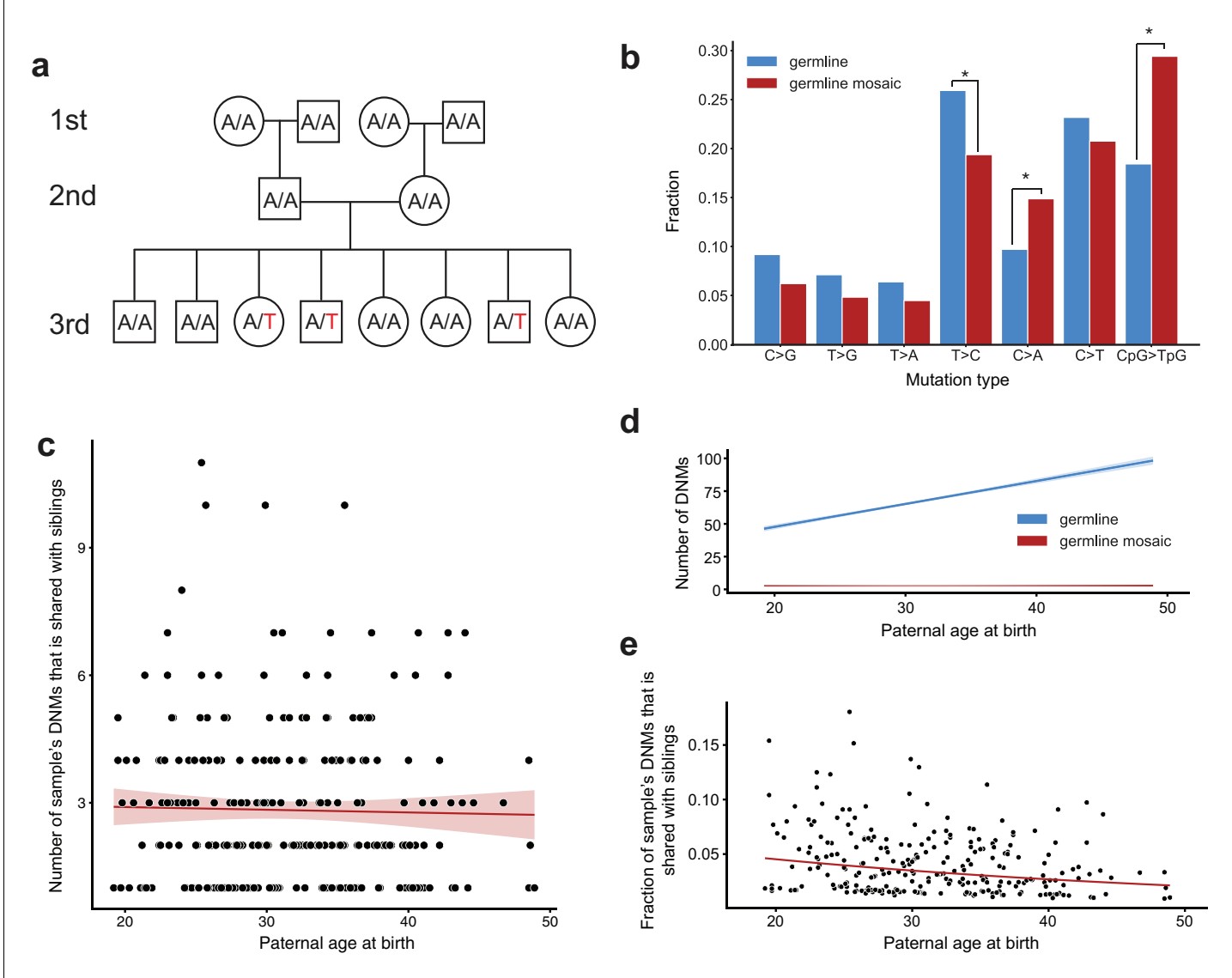

**Figure 4.** Identification of post-PGCS germline mosaicism in the second generation. (a) Mosaic variants occurring during or after primordial germ cell specification (PGCS) were defined as DNMs present in multiple third-generation siblings, and absent from progenitors in the family. (b) Comparison of mutation spectra in autosomal single-nucleotide germline mosaic variants (red, n = 288) and germline de novo variants observed in the third generation (non-shared) (blue, n = 22,644). Asterisks indicate significant differences at a false-discovery rate of 0.05 (Benjamini-Hochberg procedure), using a Chi-squared test of independence. P-values for each comparison are: C > G: 6.84e-2, T > G: 0.169, T > A: 0.236, T > C: 1.51e-2, C > A: 4.31e-3, C > T: 0.385, CpG >TpG: 2.26e-6. (c) For each third-generation individual, we calculated the number of their DNMs that was shared with at least one sibling, and plotted this number against the individual's paternal age at birth. The red line shows a Poisson regression (identity link) predicting the mosaic number as a function of paternal age at birth. (d) We fit a Poisson regression predicting the total number of germline single-nucleotide DNMs observed in the third-generation individuals as a function of paternal age at birth, and plotted the regression line (with 95% CI) in blue. In red, we plotted the line of best fit (with 95% CI) produced by the regression detailed in (c). (e) For each third-generation individual, we divided the number of their DNMs that occurred during or post-PGCS in a parent (i.e., that were shared with a sibling) by their total number of DNMs (germline +germline mosaic), and plotted this fraction of shared germline mosaic DNMs against their paternal age at birth.

DOI: https://doi.org/10.7554/eLife.46922.012

parent's germline, confirming that these variants comprise a non-negligible fraction of all de novo germline mutations.

The mutation spectrum for non-shared germline de novo variants was significantly different than the spectrum for shared germline mosaic variants (*Figure 4b*). Specifically, we found enrichments of CpG >TpG and C > A mutations, and a depletion of T > C mutations, in shared germline mosaic

variants when compared to all unshared germline de novo variants observed in the third generation (*Figure 4b*). An enrichment of CpG >TpG mutations in germline mosaic DNMs, which was also seen in a recent report on mutations shared between siblings (*Jónsson et al., 2018*), is particularly intriguing, as many C > T transitions in a CG dinucleotide context are thought to occur due to spontaneous deamination of methylated cytosine (*Fryxell and Zuckerkandl, 2000*). Indeed, DNA methylation patterns are highly dynamic during gametogenesis; evidence in mouse demonstrates that the early primordial germ cells are highly methylated, but experience a global loss of methylation during expansion and migration to the genital ridge, followed by a re-establishment of epigenetic marks (at different time points in males and females) (*Seisenberger et al., 2012*; *Reik et al., 2001*).

We also tabulated the number of each third-generation individual's DNMs that was shared with one or more of their siblings. As reported in the recent analysis of germline mosaicism (*Jónsson et al., 2018*), we observed that the number of shared germline mosaic DNMs does not increase with paternal age (p=0.647, *Figure 4c*, Materials and methods). Thus, a de novo mutation sampled from the child of a younger father is more likely to recur in a future child, as early-occurring, potentially mosaic mutations comprise a larger proportion of all DNMs present among the younger father's sperm population (*Figure 4d*). Conversely, a de novo mutation sampled from the child of an older father is less likely to recur, as the vast majority of DNMs in that father's gametes will have arisen later in life in individual spermatogonial stem cells (*Figure 4d*) (*Campbell et al., 2014a*; *Jónsson et al., 2018*). Consistent with this expectation, we observed a significant age-related decrease in the proportion of shared germline mosaic DNMs (p=1.61e-5, *Figure 4e*). Although families with large numbers of siblings are expected to offer greater power to detect shared, germline mosaic DNMs, we verified that neither the mosaic fraction nor the number of mosaic DNMs observed in third-generation children are significantly associated with the number of siblings in a family (Materials and methods).

## Identifying gonosomal mosaicism in the second generation

We further distinguished germline mosaicism from mutations that occurred before primordial germ cell specification, but likely following the fertilization of second-generation zygotes. De novo mutations that occur prior to PGCS can be present in both blood and germ cells; we therefore sought to characterize these 'gonosomal' variants that likely occurred early during the early post-zygotic development of second-generation individuals (*Besenbacher et al., 2015*; *Campbell et al., 2015*; *Campbell et al., 2014a*; *Campbell et al., 2014b*; *Rahbari et al., 2016*; *Harland et al., 2017*; *Jónsson et al., 2018*). We assumed that these gonosomal mutations would be genotyped as heterozygous in a second-generation individual, but exhibit a distinct pattern of 'incomplete linkage' to informative heterozygous alleles nearby (Materials and methods, *Figure 5a*) (*Feusier et al., 2018*; *Harland et al., 2017*; *Jónsson et al., 2018*). If these variants occurred early in development, and were present in both the blood and germ cells, we could also validate them by identifying third-generation individuals that inherited the variants with a balanced number of reads supporting the reference and alternate alleles (*Figure 5a*).

In total, we identified 475 putative autosomal gonosomal DNMs, which were also validated by visual inspection (*Supplementary file 5*). In contrast to single-gamete germline DNMs observed in the second-generation, gonosomal mutations appeared to be sex-balanced with respect to the parental haplotype on which they occurred; 52% (249/475) of all gonosomal DNMs occurred on a paternal haplotype, as compared to ~80% of germline DNMs observed in the second generation. Similarly, no significant enrichment of particular gonosomal mutation types was observed on either parental haplotype at a false discovery rate of 0.05 (*Figure 5b*), though we found that T > A transversions are enriched in gonosomal DNMs when compared to single-gamete germline DNMs observed in the second generation (p=2.32e-3) (*Figure 5c*). Unlike single-gamete germline DNMs, there were no significant effects of parental age on gonosomal DNM counts (maternal age, p=0.132; paternal age, p=0.225) (*Figure 5d*). However, a recent study found tentative evidence for a maternal age effect on de novo mutations that arise in the early stages of zygote development (*Gao et al., 2019*). As noted in this previous study, we are likely underpowered to detect a possible maternal age effect using the numbers of second-generation individuals in the CEPH/Utah dataset. Overall, our results demonstrate that over 9% (475/5,017) of all candidate autosomal germline mutations observed in the second generation were, in fact, post-zygotic in these second-generation individuals. Perhaps most importantly, approximately 6% of candidate de novo mutations detected in

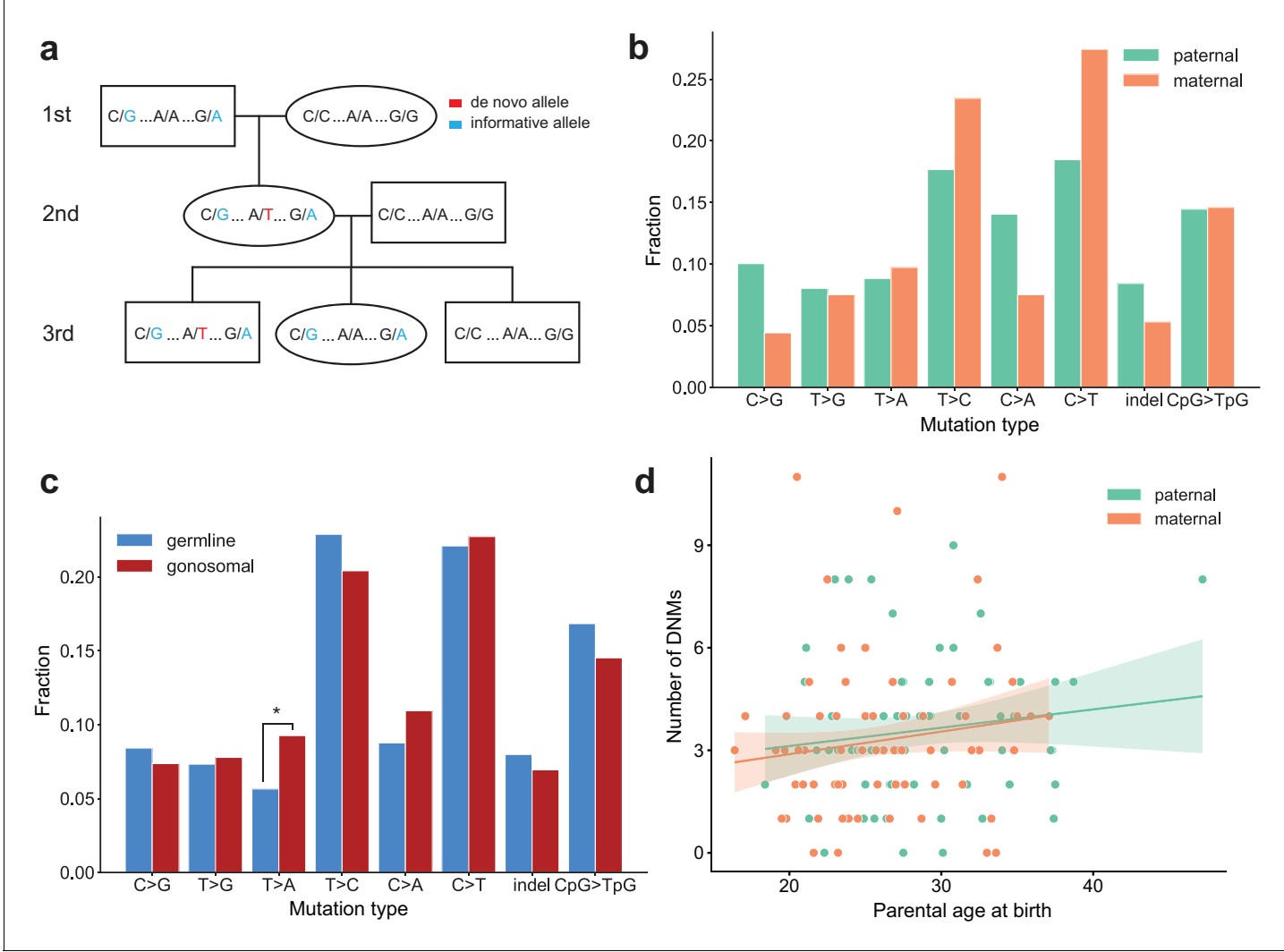

**Figure 5.** Identification of gonosomal mutations in the second generation. (**a**) Gonosomal post-zygotic variants were identified as DNMs in a second-generation individual that were inherited by one or more third-generation individuals, but exhibited incomplete linkage to informative heterozygous sites nearby. (**b**) Comparison of mutation spectra in single-nucleotide gonosomal DNMs that occurred on the paternal (n = 249) or maternal (n = 226) haplotypes. No significant differences were found at a false-discovery rate of 0.05 (Benjamini-Hochberg procedure), using a Chi-squared test of independence. P-values for each comparison are: C > G: 3.05e-2, T > G: 0.972, T > A: 0.858, T > C: 0.148, C > A: 3.31e-2, C > T: 2.66e-2, indel: 0.247, CpG >TpG: 0.932. (**c**) Comparison of mutation spectra in autosomal single-nucleotide germline DNMs observed in the second-generation (non-gonosomal) (n = 4,542) and putative gonosomal mutations (n = 475) in the second generation. Asterisks indicate significant differences at a false-discovery rate of 0.05 (Benjamini-Hochberg procedure), using a Chi-squared test of independence. P-values for each comparison are: C > G: 0.517, T > G: 0.800, T > A: 2.32e-3, T > C: 0.255, C > A: 0.129, C > T: 0.805, indel: 0.446, CpG >TpG: 0.212. (**d**) Numbers of phased gonosomal variants as a function of parental age at birth. Poisson regressions (with 95% confidence bands) were fit for the mutations phased to the maternal and paternal haplotypes separately using an identity link. A diagram of an identification strategy for post-zygotic gonosomal DNMs (using only two generations) is presented in *Figure 5—figure supplement 1*.

DOI: https://doi.org/10.7554/eLife.46922.013

The following figure supplement is available for figure 5:

**Figure supplement 1.** Strategy for identifying post-zygotic DNMs using two generations.

DOI: https://doi.org/10.7554/eLife.46922.014

the second generation with an allele balance >= 0.2 (303/5,017) were determined to be post-zygotic, and present in both somatic and germ cells. This suggests that a fraction of many germline de novo mutation datasets are comprised of truly post-zygotic DNMs, rather than mutations that occurred in a single parental gamete.

We note that our analysis pipeline may erroneously classify some gonosomal and shared germline mosaic DNMs. Namely, our count of gonosomal DNMs may be an underestimate, since our requirement that the second-generation individual be heterozygous precludes the detection of post-zygotic mosaic mutations at very low frequency in blood. Also, blood cells represent only a fraction of the total somatic cell population, and we cannot rule out the possibility that mosaicism apparently restricted to the germline may, in fact, be present in other somatic cells that were not sampled in this study (*Biesecker and Spinner, 2013*).

## Discussion

Using a cohort of large, multi-generational CEPH/Utah families, we identified a high-confidence set of germline de novo mutations that were validated by transmission to the following generation. We determined the parental gamete-of-origin for nearly all of these DNMs observed in the second generation and produced estimates of the maternal and paternal age effects on the number of DNMs in offspring. Then, by comparing parental age effects among pedigrees with large third generations whose birth dates span as many as 27 years, we found that families significantly differed with respect to these age effects. Finally, we identified gonosomal and shared germline mosaic de novo variants which appear to differ from single-gamete germline DNMs with respect to mutational spectra and magnitude of the sex bias.

Understanding family differences in both mutation rates and parental age effects could enable the identification of developmental, genetic, and environmental factors that impact this variability. The fact that there were detectable differences in parental age effects between families is striking in light of the fact that the CEPH/Utah pedigrees comprise mostly healthy individuals, and that at the time of collection they resided within a relatively narrow geographic area (*Malhotra et al., 2005*; *Dausset et al., 1990*). We therefore suspect that our results understate the true extent of variability in mutation rates and age effects among families with diverse inherited risk for mutation accumulation, and who experience a wide range of exposures, diets, and other environmental factors. Supporting this hypothesis, a recent report identified substantial differences in the mutation spectra of variants in populations of varied ancestries, suggesting that genetic modifiers of the mutation rate may exist in humans, as well as possible differences in environmental exposures (*Harris and Pritchard, 2017*; *Mathieson and Reich, 2017*). Another explanation (that we are unable to explore) for the range of de novo mutation counts in firstborn children across families is variability in the age at which parents enter puberty. For example, a father entering puberty at an older age could result in less elapsed time between the start of spermatogenesis and the fertilization of his first child's embryo. Compared to another male parent of the same age, his sperm will have accumulated fewer mutations by the time of conception. Of course, this hypothesis assumes that for both fathers, three parameters are identical: the mutation rate at puberty, the yearly mutation rate increase following puberty, and age at fertilization of the first child's embryo. Moreover, we note that replication errors are unlikely to be the sole source of de novo germline mutations (*Gao et al., 2019*). Overall, the potential sources of inter-family variability in mutation rates remain mysterious, and we anticipate that future studies will be needed to uncover the biological underpinnings of this variability.

Our observation of germline mosaicism, a result of de novo mutations that occur during or post-PGCS, has broad implications for the study of human disease and estimates of recurrence risks within families (*Jónsson et al., 2018*; *Campbell et al., 2014b*; *Biesecker and Spinner, 2013*; *Forsberg et al., 2017*; *Krupp et al., 2017*). If a de novo mutation is found to underlie a genetic disorder in a child, it is critical to understand the risk of mutation recurrence in future offspring. We estimate that ~3% of germline de novo mutations originated as a mosaic in the germ cells of a parent. This result corroborates recent reports (*Rahbari et al., 2016*; *Jónsson et al., 2018*) and demonstrates that a substantial fraction of all germline DNMs may be recurrent within a family. We also find that the mutation spectrum of shared germline mosaic DNMs is significantly different than the spectrum for single-gamete germline DNMs, raising the intriguing possibility that different mechanisms contribute to de novo mutation accumulation throughout the proliferation of primordial germ cells and later stages of gametogenesis. For instance, the substantial epigenetic reprogramming that occurs following primordial germ cell specification may predispose cells at particular developmental time points to certain classes of de novo mutations, such as C > T transitions at CpG dinucleotide sites (*Gao et al., 2019*).

Recurrent DNMs across siblings can also manifest as a consequence of gonosomal mosaicism in parents (*Biesecker and Spinner, 2013*; *Jónsson et al., 2018*). Although it can be difficult to distinguish gonosomal mosaicism from both single-gamete germline de novo mutation and germline mosaicism, we have identified a set of putative gonosomal mosaic mutations that are sex-balanced with respect to the parental haplotype on which they occurred, and do not exhibit any detectable dependence on parental age at birth. Both of these observations are expected if gonosomal mutations arise after zygote fertilization, rather than during the process of gametogenesis. We do, however, find that T > A transversions are enriched in gonosomal DNMs, as compared to DNMs that occurred exclusively in the germline of a parent. Overall, we estimate that approximately 10% of candidate germline de novo mutations in our study were, in fact, gonosomal mutations that occurred during the early cell divisions of the offspring, rather than in a single parental gamete. Prior work in cattle has estimated the fraction of mosaic DNMs that occur during early cell divisions to be even higher, suggesting that these mosaic mutations make up a large fraction of DNMs that are reported to have occurred in a single parental gamete (*Harland et al., 2017*).

These results underscore the power of large, multi-generational pedigrees for the study of de novo human mutation and yield new insight into the mutation dynamics that exist due to factors such as parental age and sex, as well as family of origin. Given that we studied only 33 large pedigrees, the mutation rate variability we observe is very likely an underestimate of the full range of variability worldwide. We therefore anticipate future studies of multi-generational pedigrees that will help to dissect the relative contributions of genetic background, developmental timing, and myriad environmental factors.

# Materials and methods

## Key resources table

| Reagent type (species) or resource | Designation | Source or reference | Identifiers | Additional information |
|---|---|---|---|---|
| Software, algorithm | Genome Analysis Toolkit (GATK) | *DePristo et al., 2011* | v3.5.0; RRID: SCR_001876 | |
| Software, algorithm | peddy | *Pedersen and Quinlan, 2017a* | v0.4.3; RRID: SCR_017287 | |
| Software, algorithm | cyvcf2 | *Pedersen and Quinlan, 2017b* | v0.11.2 | |
| Software, algorithm | mosdepth | *Pedersen and Quinlan, 2018* | v0.2.4 | |
| Software, algorithm | pysam | https://github.com/pysam-developers/pysam | v0.15.2 | |
| Software, algorithm | python | https://www.python.org/ | v3.7.3; RRID: SCR_008394 | |
| Software, algorithm | R | https://www.r-project.org/ | v3.4.4; RRID: SCR_001905 | |
| Software, algorithm | Integrative Genomics Viewer (IGV) | *Thorvaldsdóttir et al., 2013* | v2.4.11; RRID: SCR_011793 | |
| Software, algorithm | samtools | *Li et al., 2009* | RRID: SCR_002105 | |
| Software, algorithm | BWA-MEM | *Li, 2013* | v0.7.15; RRID: SCR_010910 | |

## Genome sequencing

Whole-genome DNA sequencing libraries were constructed with 500 ng of genomic DNA isolated from blood, utilizing the KAPA HTP Library Prep Kit (KAPA Biosystems, Boston, MA) on the SciClone NGS instrument (Perkin Elmer, Waltham, MA) targeting 350 bp inserts. Post-fragmentation (Covaris, Woburn, MA), the genomic DNA was size selected with AMPure XP beads using a 0.6x/0.8x ratio.

The libraries were PCR amplified with KAPA HiFi for 4–6 cycles (KAPA Biosystems, Boston, MA). The final libraries were purified with two 0.7x AMPureXP bead cleanups. The concentration of each library was accurately determined through qPCR (KAPA Biosystems, Boston, MA). Twenty four libraries were pooled and loaded across four lanes of a HiSeqX flow cell to ensure that the libraries within the pool were equally balanced. The final pool of balanced libraries was loaded over an additional 16 lanes of the Illumina HiSeqX (Illumina, San Diego, CA). 2 × 150 paired-end sequence data was generated. This efficient pooling scheme targeted ~30X coverage for each sample.

## DNA sequence alignment

Sequence reads were aligned to the GRCh37 reference genome (including decoy sequences from the GATK resource bundle) using BWA-MEM v0.7.15 (*Li, 2013*). The aligned BAM files produced by BWA-MEM were de-duplicated with samblaster (*Faust and Hall, 2014*). Realignment for regions containing potential short insertions and deletions and base quality score recalibration was performed using GATK v3.5.0 (*DePristo et al., 2011*). Alignment quality metrics were calculated by running samtools 'stats' and 'flagstats' (*Li et al., 2009*) on aligned and polished BAM files.

## Variant calling

Single-nucleotide and short insertion/deletion variant calling was performed with GATK v3.5.0 (*DePristo et al., 2011*) to produce gVCF files for each sample. Sample gVCF files were then jointly genotyped to produce a multi-sample project level VCF file.

## Sample quality control and filtering

We used peddy (*Pedersen and Quinlan, 2017a*) to perform relatedness and sample sequencing quality checks on all CEPH/Utah samples. We discovered a total of 10 samples with excess levels of heterozygosity (proportion of heterozygous calls > 0.2). Many of these samples were also listed as being duplicates of other samples in the cohort, indicating possible sample contamination prior to sequencing. We therefore removed all 10 samples with a heterozygous genotype proportion exceeding 0.2 from further analysis. In total, we were left with 593 first-, second-, and third-generation samples with high-quality sequencing data.

## Identifying DNM candidates

We identified high-confidence de novo mutations from the joint-called VCF in the second and third generations as follows, using cyvcf2 (*Pedersen and Quinlan, 2017b*). For each variant, we required that the child possessed a unique genotyped allele absent from both parents; when identifying de novo variants on the X chromosome, we required male offspring genotypes to be homozygous. We required the aligned sequencing depth in the child and both parents to be >= 12 reads, Phred-scaled genotype quality (GQ) to be >= 20 in the child and both parents, and no reads supporting the de novo allele in either parent. We removed de novo variants within low-complexity regions (*Li, 2014*; *Turner et al., 2017*), and any variants that were not listed as 'PASS' variants by GATK HaplotypeCaller. Finally, we removed DNMs with likely DNM carriers in the cohort; we define carriers as samples that possess the DNM allele, other than the sample with the putative DNM and his/her immediate family (i.e., siblings, parents, or grandparents). We adapted a previously published strategy (*Jónsson et al., 2017*) to discriminate between 'possible carriers' of the DNM allele (samples genotyped as possessing the de novo allele), and 'likely carriers' (a subset of 'possible carriers' with depth >= 12, allele balance >= 0.2, and Phred-scaled genotype quality >= 20). We removed putative DNMs for which there were any 'likely carriers' of the allele in the cohort. We then separated the candidate variants observed in the second-generation into true and false positives based on transmission to the third generation. For each candidate second-generation variant, we assessed whether the DNM was inherited by at least one member of the third generation; to limit our identification of false positive transmission events, we required third-generation individuals with inherited DNMs to have a depth >= 12 reads at the site and Phred-scaled genotype quality >= 20. We defined 'transmitted' second-generation variants as variants for which the median allele balance across transmissions was >= 0.3. One CEPH/Utah family (family ID 26) contains only four sequenced grandchildren (*Supplementary file 1*); therefore, we did not include the two second-generation

individuals from this family in our analysis of DNMs observed in the second-generation, as we lacked power to detect high-quality transmission events.

Because we were unable to validate DNMs observed in the third generation by transmission, we applied a more stringent set of quality filters to all third-generation DNMs. We required the same filters as applied to all second-generation DNMs, but additionally required that the allele balance in each DNM was >= 0.3. We further required that there were no possible carriers of the de novo allele in the rest of the cohort. For each DNM in the third generation, we assessed if any of the third-generation individuals' grandparents were genotyped as possessing the DNM allele; if so, we removed that DNM from further analysis (see section entitled 'Estimating a missed heterozygote rate'). Finally, we removed a total of 319 candidate germline DNMs in the third generation after finding evidence that these were, in fact, post-zygotic mutations (see section entitled 'Identifying gonosomal mutations').

## Determining the parent of origin for single-gamete germline DNMs

To determine the parent of origin for each de novo variant in the second generation, we phased mutation alleles by transmission to a third generation, a technique which has been described previously (*Jónsson et al., 2017*; *Kong et al., 2012*; *Goldmann et al., 2016*; *Rahbari et al., 2016*) (*Figure 1—figure supplement 2a*). We searched 200 kbp upstream and downstream of each DNM for informative variants, defined as alleles present as a heterozygote in the second-generation individual, observed in only one of the two parents, and observed in each of the third-generation individuals that inherited the DNM. For each of these informative variants, we confirmed that the informative variant was always transmitted with the DNM; if so, we could infer that the heterozygous variant was present on the same haplotype as the DNM (assuming recombination did not occur between the DNM and the flanking informative variants), and assign the first-generation parent with the informative variant as the parent of origin (*Figure 1—figure supplement 2a*). For each second-generation DNM, we identified all transmission patterns (i.e., combinations of a first-generation parent, second-generation child, and set of third-generation grandchildren that inherited both the informative variant and the DNM). We only assigned a confident parent-of-origin at sites where the most frequent transmission pattern occurred at >= 75% of all informative sites.

We additionally phased de novo variants in the second generation, as well as all DNMs in the third generation, using 'read tracing' (also known as 'read-backed phasing') (*Jónsson et al., 2017*; *Goldmann et al., 2016*). Briefly, for each de novo variant, we first searched for nearby (within one read fragment length, 500 bp) variants present in the proband and one of the two parents. Thus, if the de novo variant was present on the same read as the inherited variant, we could infer haplotype sharing, and determine that the de novo event occurred on that parent's chromosome (*Figure 1—figure supplement 2b*). Similarly, if the de novo variant was not present on the same read as the inherited variant, we could infer that the de novo event occurred on the other parent's chromosome.

We were also able to determine the parent-of-origin for many of the shared germline mosaic variants by leveraging haplotype sharing across three generations (*Jónsson et al., 2018*). If all third-generation individuals with a post-PGCS DNM shared a haplotype with a particular first-generation grandparent, we assigned that first-generation grandparent's child (i.e., one of the two second-generation parents) as the parent of origin.

In the second generation, the read tracing and haplotyping sharing phasing strategies were highly concordant, and the parent-of-origin predictions agreed at 98.8% (969/980) of all DNMs for which both strategies could be applied.

## Calculating the rate of germline mutation

Given the filters we employed to identify high-confidence de novo mutations, we needed to calculate the fraction of the genome that was considered in our analysis. To this end, we used mosdepth (*Pedersen and Quinlan, 2018*) to calculate per-base genome coverage in all CEPH/Utah samples, excluding low-complexity regions (*Li, 2014*) and reads with mapping quality <20 (the minimum mapping quality threshold used by GATK HaplotypeCaller in this analysis). For each second- and third-generation child, we then calculated the number of all genomic positions that had at least 12 aligned sequence reads in the child's, mother's, and father's genome (excluding the X chromosome). In the

second generation, the median number of callable autosomal base pairs per sample was 2,582,336,232. For each individual, we then divided their count of autosomal de novo mutations by the resulting number of base pairs, and divided the result by two to obtain a diploid human mutation rate per base pair per generation. The median second-generation germline SNV mutation rate was calculated to be $1.143 \times 10^{-8}$ per base pair per generation. We then adjusted this mutation rate based on our estimated false positive rate (FPR) and our estimated 'missed heterozygote rate' (MHR; see section entitled 'Estimating a missed heterozygote rate') as follows:

```
adj_mu = mu * (1 – FPR/1 – MHR)
adj_mu = 1.143e-8 * (1–0.045/1–0.004)
```

## Assessing age effect variability between families

Using the full call set of de novo variants in the third generation (excluding the recurrent, post-PGCS DNMs and likely post-zygotic DNMs) we first fit a simple Poisson regression model that calculated the effect of paternal age on total autosomal DNM counts in the R statistical language (v3.5.1) as follows:

```
glm(autosomal_dnms ~ dad_age, family = poisson(link='identity'))
```

This model returned a highly significant effect of paternal age on total DNM counts (1.72 DNMs per year of paternal age, p<2e-16), but was agnostic to the family from which each third-generation individual was 'sampled.' Importantly, a number of third-generation individuals in the CEPH/Utah cohort share grandparents, and may therefore be considered members of the same family, despite having unique second-generation parents (*Figure 3—figure supplement 1*). For all subsequent analysis, we defined a 'family' as the unique group of two second-generation parents and their third-generation offspring (*Figure 3—figure supplement 1*). In the CEPH/Utah cohort, there are a total of 40 'families' meeting this definition.

To test for significant variability in paternal age effects between families, we fit the following model:

```
glm(autosomal_dnms ~ dad_age * family_id,
family = poisson(link='identity'))
```

Which can also be written in an expanded form as:

```
glm(autosomal_dnms ~ dad_age + family_id + dad_age:family_id,
family = poisson(link='identity'))
```

To assess the significance of each term in the fitted model, we performed an analysis of variance (ANOVA) as follows:

```
m = glm(autosomal_dnms ~ dad_age + family_id + dad_age:family_id, family = poisson
(link='identity'))
anova(m, test='Chisq')
```

The results of this ANOVA are shown in *Appendix 1—Table 1*. In summary, this model contained the fixed effect of paternal age, as well as different regression intercepts within each 'grouping factor' (i.e., family ID). Additionally, this model includes an interaction between paternal age and family ID, allowing for the effect of paternal age (i.e., the slope of the regression) to vary within each grouping factor.

To account for variable sequencing coverage across CEPH/Utah samples, we additionally calculated the callable autosomal fraction for all third-generation individuals by summing the total number of nucleotides covered by >= 12 reads in the third-generation individual and both of their second-generation parents, excluding low-complexity regions and reads with mapping quality <20 (see section entitled 'Calculating the rate of germline mutation').

Since we only consider the effect of paternal age on the mutation rate, we can model the mutation rate (mu) as:

```
mu = Bp * Ap +B0
```

Where Bp is the paternal age effect, Ap is the paternal age, and B0 is an intercept term.

Therefore, the number of DNMs in a sample is assumed to follow a Poisson distribution, with the expected mean of the distribution defined as:

```
E(# DNMs) = mu * callable_fraction
E(# DNMs) = (Bp * Ap + B0) * callable_fraction
E(# DNMs) = (Bp * Ap * callable_fraction) + (callable_fraction * B0)
```

As our analysis only considers the effect of paternal age on total DNM counts, we can thus scale Ap (paternal age at birth) by the `callable_fraction`, generating a term called `dad_age_scaled`, and fit the following model, which takes each sample's callable fraction into account:

```
glm(autosomal_dnms  ~  dad_age_scaled  +  autosomal_callable_fraction  +0,
family = poisson(link='identity'))
```

Then, we can determine whether inter-family differences remain significant by comparing the above null model to a model that takes family into account:

```
glm(autosomal_dnms                  ~                  dad_age_scaled              *
family_id + autosomal_callable_fraction + 0, family = poisson(link='identity'))
```

After running an ANOVA to compare the two models, we find that the model incorporating family ID is a significantly better fit (ANOVA: p=9.359e-10).

We previously identified significant effects of both maternal and paternal age on DNM counts (*Figure 2a*). Therefore, to account for the non-negligible effect of maternal age on DNM counts, we fit a final model that incorporated the effects of both maternal and paternal age, as well as family ID, on total DNM counts as follows:

```
glm(autosomal_dnms  ~  dad_age  +mom_age  +family_id,  family  =  poisson
(link='identity'))
```

We then performed an ANOVA on the model, and found that a model incorporating a family term is a significantly better fit than a model that includes the effects of paternal and maternal age alone (p=2.12e-5).

## Identifying post-PGCS mosaic mutations

To identify post-PGCS mosaic variants, we searched the previously generated callset of single-nucleotide DNMs in the third generation ('Identifying DNM candidates') for de novo single-nucleotide mutations that appeared in two or more third-generation siblings. As a result, all filters applied to the germline third-generation DNM callset were also applied to the post-PGCS mosaic variants. We validated all putative post-PGCS mosaic variants by visual inspection using the Integrative Genomics Viewer (IGV) (*Thorvaldsdóttir et al., 2013*). In a small number of cases (32), we found evidence for the post-PGCS mosaic variant in one of the two second-generation parents. Reads supporting the post-PGCS mosaic variant were likely filtered from the joint-called CEPH/Utah VCF output following local re-assembly with GATK, though they are clearly present in the raw BAM alignment files. We removed these 32 variants, at which an second-generation parent possessed two or more reads of support for the mosaic DNM allele in the aligned sequencing reads.

## Assessing age effects on post-PGCS DNMs

To identify a paternal age effect on the number of post-PGCS DNMs transmitted to third-generation children, we tabulated the number of each third-generation individual's DNMs that was shared with at least one of their siblings. We then fit a Poisson regression as follows, regressing the number of mosaic DNMs in each third-generation individual against their father's age at birth:

```
glm(mosaic_number ~dad_age, family = poisson(link='identity'))
```

We did not find a significant effect of paternal age (p=0.647).

Using the predicted paternal age effects on germline DNM counts and post-PGCS DNM counts, we determined that the fraction of post-PGCS DNMs should decrease non-linearly with paternal age (*Figure 4e*). Therefore, to assess the effect of paternal age on the fraction of each third-generation individual's DNMs that occurred post-PGCS in a parent, we fit the following model:

```
lm(log(mosaic_fraction) ~dad_age)
```

We found a significant effect of paternal age on the post-PGCS mosaic fraction (p=1.61e-5).

As we may be more likely to identify shared, post-PGCS DNMs in families with larger numbers of third-generation siblings, we additionally tested whether the fraction of post-PGCS DNMs in each child was dependent on the number of their siblings in the family by performing a correlation test as follows:

```
cor.test(mosaic_fraction, n_siblings)
```

We did not observe a significant correlation between a third-generation individual's number of siblings and the fraction of their DNMs that was shared with a sibling (p=0.882). We also did not observe a significant correlation between a third-generation individual's number of siblings and the total number of their DNMs shared with a sibling (p=0.426).

## Identifying gonosomal mutations

To identify variants that occurred early in post-zygotic development, we identified de novo single-nucleotide variants in the second generation using the same genotype quality and population-based filters as described previously ('Identifying DNM candidates'). Then, to distinguish single-gamete germline de novo mutations from post-zygotic DNMs (de novo mutations that occurred in the cell divisions following fertilization of the second-generation individual's embryo), we employed a previously described method (*Harland et al., 2017*; *Feusier et al., 2018*; *Jónsson et al., 2018*) that relies on linkage between DNMs and informative heterozygous alleles nearby. In this approach, which is similar in principle to the strategy used for phasing germline second-generation DNMs, we first search ±200 kbp up- and down-stream of the de novo allele in the second-generation individual for 'informative' alleles; that is, alleles that are present in only one first-generation parent, and inherited by the second-generation child (*Figure 5a*). Then, we identify all of the third-generation grandchildren that inherited the informative alleles. If all of the third-generation individuals that inherited the informative alleles also inherited the DNM, we infer that the DNM occurred in the germline of the first-generation parent with the informative allele. However, if one or more third-generation individuals inherited the informative alleles but did *not* inherit the DNM, we can infer that the DNM occurred sometime following the fertilization of the second-generation sample's embryo. This is because the DNM is not always present on the background haplotype that the second-generation individual inherited from their informative first-generation parent. Using this approach, we do not apply any allele balance filters to putative gonosomal DNMs in the second generation, instead relying on linkage to distinguish them from germline DNMs. As with germline de novo mutations observed in the second-generation, to limit our identification of false positive events, we required third-generation individuals with inherited DNMs to have a depth >= 12 reads at the site, Phred-scaled genotype quality (GQ) >= 20, and for the median allele balance across transmissions to be >= 0.3.

Additionally, we can use an orthogonal method to distinguish single-gamete germline DNMs from post-zygotic DNMs. In this second approach, we identify all heterozygous sites ± 500 base pairs

(approximately one read length) from a DNM in a child. Then, by assessing the linkage of the DNM and heterozygous alleles, we look for evidence of three distinct haplotypes in the child (*Figure 5— figure supplement 1*). If we observe at least two reads supporting a third haplotype (i.e., reads that indicate incomplete linkage between the DNM and the informative heterozygous allele), we inferred that the DNM occurred post-zygotically in the child. We applied this method to all putative germline DNMs identified in the third generation, and discovered that 319 of apparent germline DNMs showed evidence of being post-zygotic mutations that occurred following the fertilization of the third-generation embryo. We removed these DNMs from all analyses of third-generation germline DNMs.

We validated all putative gonosomal variants in the second generation by visual inspection using the Integrative Genomics Viewer (IGV) (*Thorvaldsdóttir et al., 2013*).

## Estimating a 'missed heterozygote rate' for DNM detection

Infrequently, variant calling methods such as GATK may incorrectly assign genotypes to samples at particular sites in the genome. When identifying de novo variants, we require that children possess genotyped alleles that are absent from either parent; thus, genotyping errors in parents could lead us to assign variants as being de novo, when in fact one or both parents possessed the variant and transmitted the allele. Given the multi-generational structure of our study cohort, we were able to estimate the rate at which our variant calling and filtering pipeline mis-genotyped a second-generation parent as being homozygous for a reference allele. To estimate this 'missed heterozygote' rate in our dataset, we looked for any cases in which one or more third-generation individuals possessed a putative de novo variant (i.e. possessed an allele absent from both second-generation parents). Then, we looked at the sample's grandparental (first-generation) genotypes for evidence of the same variant. If one or more grandparents was genotyped as having high-quality evidence for the de novo allele (depth >= 12 and Phred-scaled genotype quality >= 20), we inferred that the variant could have been 'missed' in the second generation, despite being truly inherited. We estimate the missed heterozygote rate (MHR) to be 0.4%, by dividing the total number of third-generation DNMs with grandparental support by the total number of third-generation DNMs (100/25,075). In a small number of CEPH/Utah pedigrees, some members of the first-generation (grandparental) generation were not sequenced (6 grandparents in five families, *Supplementary file 1*). As a result these families are underpowered to detect evidence of third-generation DNM alleles in the first generation, and our MHR is likely a slight underestimate.

## Estimating a false positive rate for de novo mutation detection

In a separate set of sequencing runs, a total of 8 first-generation grandparents were re-sequenced to a greater genome-wide median depth of 60X (*Figure 1—figure supplement 1d*). However, when variant calling and joint genotyping was performed on all 603 CEPH/Utah samples, the 30X data for these grandparents was used. Therefore, we sought to estimate the false positive rate for our de novo mutation detection strategy using the de novo mutation calls in the children of these eight first-generation individuals. For each of the children (second-generation) of these high-coverage first-generation individuals, we looked for evidence of the second-generation DNMs in the 60X alignments from their parents. Specifically, for each second-generation DNM, we counted the number of reads supporting the DNM allele in each of the first-generation parents, excluding reads with mapping quality <20 (the minimum mapping quality imposed by GATK HaplotypeCaller in our analysis), and excluding bases with base qualities < 20 (the minimum base quality imposed by GATK HaplotypeCaller in our analysis). If we observed two or more reads supporting the second-generation DNM in a first-generation parent's 60X alignments, we considered the second-generation DNM to be a false positive. Of the 202 de novo mutations called in the four second-generation children of the high-coverage first-generation parents, we find nine mutations with at least two reads of supporting evidence in the 60X first-generation alignments. Thus, we estimate our false positive rate for de novo mutation detection to be approximately 4.5% (9/202).

## Data and code availability

Code used for statistical analysis and figure generation has been deposited on GitHub as a collection of annotated Jupyter Notebooks: https://github.com/quinlan-lab/ceph-dnm-

manuscript (*Sasani, 2019*; copy archived at https://github.com/elifesciences-publications/ceph-dnm-manuscript/blob/master/README.md). Data files containing high-confidence de novo mutations, as well as the gonosomal and post-primordial germ cell specification (PGCS) mosaic mutations, are included with these Notebooks. To mitigate compatibility issues, we have also made all notebooks available in a Binder environment, accessible at the above GitHub repository (*Sasani, 2019*).

# Acknowledgements

We thank all of the Utah individuals who participated in the CEPH consortium. We also thank Ray White, Jean-Marc Lalouel, and Mark Leppert, who were instrumental in the ascertainment of the CEPH/Utah pedigrees. We additionally thank Chad Harland and Julie Feusier for assisting our detection of post-zygotic mosaicism and Andrew Farrell for assistance with interpreting DNM calls. Finally, we thank Tim Formosa, Richard Cawthon, Amelia Wallace and many other members of the Quinlan and Jorde laboratories for insightful discussion related to the manuscript.

# Additional information

### Competing interests
Molly Przeworski: Reviewing editor, *eLife*. The other authors declare that no competing interests exist.

### Funding

| Funder | Grant reference number | Author |
|---|---|---|
| National Institute of General Medical Sciences | T32GM007464 | Thomas A Sasani |
| National Human Genome Research Institute | R01HG006693 | Aaron R Quinlan |
| National Human Genome Research Institute | R01HG009141 | Aaron R Quinlan |
| National Institute of General Medical Sciences | R01GM124355 | Aaron R Quinlan |
| National Institute of General Medical Sciences | R35GM118335 | Lynn Jorde |
| National Institute of General Medical Sciences | R01GM122975 | Molly Przeworski |

The funders had no role in study design, data collection and interpretation, or the decision to submit the work for publication.

### Author contributions
Thomas A Sasani, Data curation, Software, Formal analysis, Investigation, Visualization, Methodology, Writing—original draft; Brent S Pedersen, Software, Formal analysis, Investigation, Methodology, Writing—review and editing; Ziyue Gao, Molly Przeworski, Formal analysis, Methodology, Writing—review and editing; Lisa Baird, Resources, Data curation; Lynn B Jorde, Conceptualization, Resources, Supervision, Funding acquisition, Project administration, Writing—review and editing; Aaron R Quinlan, Conceptualization, Supervision, Funding acquisition, Writing—original draft, Project administration, Writing—review and editing

### Author ORCIDs
Thomas A Sasani (iD) https://orcid.org/0000-0003-2317-1374
Ziyue Gao (iD) http://orcid.org/0000-0001-9244-0238
Molly Przeworski (iD) http://orcid.org/0000-0002-5369-9009

### Ethics

Human subjects: Informed consent was obtained from the CEPH/Utah individuals, and the University of Utah Institutional Review Board approved the study (University of Utah IRB reference #80145).

### Decision letter and Author response

Decision letter https://doi.org/10.7554/eLife.46922.026
Author response https://doi.org/10.7554/eLife.46922.027

## Additional files

### Supplementary files

• Supplementary file 1. Pedigree structures for all CEPH/Utah families. All family and sample IDs have been anonymized, and the sexes of third-generation individuals have been hidden.
DOI: https://doi.org/10.7554/eLife.46922.015

• Supplementary file 2. IGV images of 100 randomly selected germline DNMs identified in the second generation. In each image, the first two tracks contain alignments from the first-generation parents, and the third track contains the alignments for the second-generation child. Reads with mapping quality <20 are not included, as they were not considered by our variant calling pipeline, and mismatched bases are shaded by quality score (more transparent = lower base quality).
DOI: https://doi.org/10.7554/eLife.46922.016

• Supplementary file 3. IGV images of 100 randomly selected germline. DNMs identified in the third generation In each image, the first two tracks contain alignments from the second-generation parents, and the third track contains the alignments for the third-generation child. Reads with mapping quality <20 are filtered out, as they were not considered by our variant calling pipeline, and mismatched bases are shaded by quality score (more transparent = lower base quality).
DOI: https://doi.org/10.7554/eLife.46922.017

• Supplementary file 4. IGV images of all putative post-PGCS mosaic mutations In each image, the first two tracks contain alignments from the two second-generation parents in the pedigree. All tracks below contain alignments from the third-generation children that share a DNM at the site. Reads with mapping quality <20 are filtered out, as they were not considered by our variant calling pipeline, and mismatched bases are shaded by quality score (more transparent = lower base quality).
DOI: https://doi.org/10.7554/eLife.46922.018

• Supplementary file 5. IGV images of all putative gonosomal mutations identified in the second generation. In each image, the first two, three, or four tracks contain alignments from the grandparents in the pedigree (i.e., paternal grandmother and grandfather, maternal grandmother and grandfather). In some families, one or two of the first-generation grandparents were not sequenced (see *Supplementary file 1*). The two tracks below contain alignments from the second-generation individual with the putative gonosomal mutation and that second-generation individual's spouse. The remaining tracks below contain alignments from the third-generation individuals that inherited the gonosomal mutation. Reads with mapping quality <20 are filtered out, as they were not considered by our variant calling pipeline, and mismatched bases are shaded by quality score (more transparent = lower base quality).
DOI: https://doi.org/10.7554/eLife.46922.019

• Transparent reporting form
DOI: https://doi.org/10.7554/eLife.46922.020

### Data availability

Code used for statistical analysis and figure generation has been deposited on GitHub as a collection of annotated Jupyter Notebooks: https://github.com/quinlan-lab/ceph-dnm-manuscript (copy archived at https://github.com/elifesciences-publications/ceph-dnm-manuscript). Data files containing high-confidence de novo mutations, as well as the gonosomal and post-primordial germ cell specification (PGCS) mosaic mutations, are included with these Notebooks. To mitigate compatibility issues, we have also made all notebooks available in a Binder environment, accessible at the above

GitHub repository. Aligned sequencing reads (in CRAM format) and variant calls (in VCF format) will be made available at the SRA and dbGaP under controlled access, with accession phs001872.v1.p1.

The following dataset was generated:

| Author(s) | Year | Dataset title | Dataset URL | Database and Identifier |
|---|---|---|---|---|
| Sasani TA, Pedersen BS, Gao Z, Baird L, Przeworski M, Jorde LB, Quinlan AR | 2019 | Genome sequencing of large, multigenerational CEPH/Utah families | https://www.ncbi.nlm.nih.gov/projects/gap/cgi-bin/study.cgi?study_id=phs001872.v1.p1 | NCBI dbGaP, phs001872.v1.p1 |

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

# Appendix 1

DOI: https://doi.org/10.7554/eLife.46922.021

## Supplementary Information

**Appendix 1—table 1. Results of ANOVA on fitted 'family-aware' model.**

| Term (independent variable) | DoF | Deviance | Resid. DoF | Resid. Deviance | Pr(>Chi) |
|---|---|---|---|---|---|
| dad_age | 1 | 635.77 | 348 | 502.84 | < 2.2e-16 |
| family_id | 39 | 103.43 | 309 | 399.41 | 9.667e-9 |
| dad_age:family_id | 39 | 55.34 | 270 | 344.07 | 0.04328 |

DOI: https://doi.org/10.7554/eLife.46922.022

