## [Decision Letter]

Thank you for submitting your article "Large, three-generation CEPH families reveal post-zygotic mosaicism and variability in germline mutation accumulation" for consideration by *eLife*. Your article has been reviewed by three peer reviewers, including Amy Williams as the Reviewing Editor and Reviewer #1, and the evaluation has been overseen by Mark McCarthy as the Senior Editor.

The reviewers have discussed the reviews with one another and the Reviewing Editor has drafted this decision to help you prepare a revised submission.

Summary:

Sasani et al. present a study of 40 large multi-sibling, three-generation CEPH/Utah pedigrees with the aim of estimating the rate of de novo mutations (DNMs), analyzing variation in paternal age effects, and identifying germline mosaic DNMs with their associated mutational spectra. The number of families and the fact that the CEPH/Utah pedigrees are enriched with large numbers of children in the third generation enable a more detailed study of the paternal age effect and of germline mosaic DNMs than prior studies, which have primarily analyzed trios. The finding that the paternal age effect varies by more than an order of magnitude adds to the already complex picture of mutational dynamics and provides strong evidence in support of a prior finding of a two-fold difference based on three families (Rahbari et al.). The study design enables the division of germline mosaic variants into those that seem to have arisen very early in development – gonosomal variants – and those that arise after primordial germ cell specification (post-PGCS). The mutational spectra associated with post-PGCS DNMs differ from those of other DNMs, with some possible biological explanations proposed in the text.

Essential revisions:

Overall this paper is a solid contribution to the DNM literature, but there are a few additional analyses that will help ensure the findings are robust and enhance the information already presented, as outlined below.

1) Since this manuscript's main advance regarding variation in paternal age effect over the Rahbari et al. result is greater statistical power, more robust statistical analyses of this pattern would strengthen the paper. Figure 3 presents a commendable amount of raw data in a fairly clear way, yet the authors use only a simple ANOVA to test whether different families have different dependencies on paternal age. The supplement claims that this result cannot be an artifact of low sequencing coverage because regions covered by <12 reads are excluded from the denominator, but there still might be subtle differences in variant discovery power between e.g. regions covered by 12 reads and regions covered by 30 reads. To hedge against this, the authors can define the "callable genome" continuously (point 1 under "other questions and suggestions") or they can check whether mean read coverage appears to covary with mutation rates across individuals after filtering away the regions covered by <12 reads.

2) Another concern about paternal age effects is the extent to which outlier offspring may be driving the apparent rate variation across families. If the authors were to randomly sample half of the children from each family and run the analysis again, how much is the paternal age effect rank preserved? Alternatively, how much is the family rank ordering preserved if mutations are only called from a subset of the chromosomes?

3) A factor regarding paternal age effects that is only mentioned briefly alluded to late in the paper are the differences in the intercept between families (Figure 3C). Do either the intercept or slope vary with the number of F2 children? Is there any (anti-)correlation between slope and intercept? It would seem odd if the intercept strongly impacts the slope since a low per-year rate probably should not correlate with a high initial rate at younger ages.

4) Another analysis that is informative about the cellular stage at which mutations occurred is to examine, for mutations found in F1, what fraction of these are transmitted to F2s. Putative DNMs could in fact be present in blood but not in the germline, and the study design here makes it easy to identify the fraction of such DNMs. A complexity of this is the use of multi-sample genotype calling. Perhaps dividing the genotype calling into a set called using the P0 and F1 generations only and comparing the resulting DNMs to those found in F2s (with calling in everyone) would ensure that the set of DNMs aren't biased towards those also present in the germline of F1s.

5) In the additional data files, I could not find the age of all the individuals. It would be informative if the age information can be provided on the pedigree diagrams or as a separate files with identifiers.

6) Will the sequencing data generated here be posted to the SRA or dbGaP?

7) The authors state that "a gamete sampled from a younger father is more likely to possess a DNM that will recur in a future child." This doesn't seem correct as stated – every gamete should be equally likely to possess a DNM that will recur in a future child, independent of parental age. I believe what is meant is that a particular DNM sampled from the child of a young parent is more likely to be shared with a sibling than a DNM sampled from the child of an older parent.

Other questions and suggestions:

1) One concern is in defining a site as "callable", which is of course not strictly binary. It would be good to take sequencing depth into consideration when deciding callability. Ideally this would also factor into both the FPR and MHR values. Given the three-generation study design, there is a greater opportunity to perform these analyses in a more detailed manner than in trio studies and thus to better estimate/model these rates.

2) Another factor to analyze is the use of multi-sample genotype calling and its potential to bias against the identification of non-mosaic (singleton) DNMs. Perhaps the 60x vs. 30x analysis can help estimate the rates of missing singletons in a way that is distinct from the MHR analysis.

3) What is the range of the MHR? Is there significant variation with respect to MHR among families (in cases where P0 were genotyped)? Moreover, is there any enrichment or biases in mutational spectra seen using the MHR?

4) The authors mentioned that they have removed DNMs with likely "DNM carriers" in the cohort. Does this remove DNMs where the alternate alleles are observed in only the unrelated individuals or does it also include the related individuals?

5) Other things to explore further related to parental age effects are: how do the conclusions change and/or can you detect similar variability in maternal age when analyzing phased DNMs? This may be underpowered, but for those families that share grandparents, if two brothers are in the F1 generation, do their paternal age effects differ?

6) For the parental age effect model the authors have correctly included "family-id", but for the rest of their analysis they have defined a "family" as the unique group of two F1 parents and their F2 offspring (e.g., Figure 3—figure supplement 1). Can the authors comment whether this might introduce biases in their analysis and filtering strategies, as some families are more related to each other than the rest?

7) Figure 4 shows that the number of DNMs shared with siblings does not appear to correlate with paternal age. Although it is seems unlikely to affect the result, it seems odd to report these as raw counts without correcting for the number of siblings the child has. It would be good to report the strength of the correlation between family size and shared DNM count and correct the shared counts for family size before testing for a correlation with paternal age.

8) Why, from Figure 4B, are the differences in mutational spectra found in the post-PGCS mosaic analysis only based on 289/721 of these DNMs (presumably the phased ones)?

9) A minor but important consideration here is as a term, "post-PGCS", seems to include any mutations that arise following the establishment of the germ cells, but what seems to be the intended meaning is those mutations that arise during germ cell proliferation (or related). Rewording would aid understanding here.

10) For the mutational spectra analysis of gonosomal mosaic DNMs, this and other similar analyses consider each allelic class independently. Would power increase by analyzing the data as a whole using, say, a Chi-squared six degree of freedom test?

11) The authors have applied the same filters as DNMs for identifying post-PGCS mosaic variants. They seem to have filtered candidates based on VAF > 0.2. Might this filter may be too stringent for identifying the post-PGCS events?

12) To identify gonosomal mutations the authors have applied hard VAF cut off < 0.2, considering the number of cell divisions before PGC, would this threshold be a bit too low? Would their observation change significantly if they change the threshold to <0.3?

13) What is the VAF distribution of candidate gonosomal mutations in F2? One of the filters they have used have VAF >=0.3 in F2. Might this threshold be too lax? For the gonosomal mutations that occur in F1, an expectation of a higher VAF of almost 0.5 in the F2 set seems reasonable.

14) In mosaic post-PGCS analysis: the authors have identified 32 events with supporting alleles in F1. Among these 32 mutations, do any occur in families were F1s are related? For example, do F1 19_A mom and 19_B dad share some of these mosaic mutations? If so is there any correlation in mutational burden in F1 with the age of P0?

---

## [Author Response]

Essential revisions:Overall this paper is a solid contribution to the DNM literature, but there are a few additional analyses that will help ensure the findings are robust and enhance the information already presented, as outlined below.1) Since this manuscript's main advance regarding variation in paternal age effect over the Rahbari et al. result is greater statistical power, more robust statistical analyses of this pattern would strengthen the paper. Figure 3 presents a commendable amount of raw data in a fairly clear way, yet the authors use only a simple ANOVA to test whether different families have different dependencies on paternal age. The supplement claims that this result cannot be an artifact of low sequencing coverage because regions covered by <12 reads are excluded from the denominator, but there still might be subtle differences in variant discovery power between e.g. regions covered by 12 reads and regions covered by 30 reads. To hedge against this, the authors can define the "callable genome" continuously (point 1 under "other questions and suggestions") or they can check whether mean read coverage appears to covary with mutation rates across individuals after filtering away the regions covered by <12 reads.

It is true that “callability” is not necessarily a binary quality. To address this, we have assessed whether mutation rates are correlated with mean read depth in the second- and third-generation samples. We counted the total number of sites at which all members of a trio (mother, father, and child) had depth >= 12, and then averaged the read depth across all of these sites in the child. We have included a plot of mean autosomal read depth versus autosomal mutation rates in all second- and third-generation children in Author response image 1. Overall, mutation rates do not appear to be correlated with mean read depth in the second (p = 0.92) or third-generation samples (p = 0.073), though there are a small number of third-generation samples that have both relatively low mutation rates and low mean read depths.

**Author response image 1. respfig1:** Lack of correlation between read depth and mutation rates in CEPH/Utah samples. For each second- or third-generation CEPH/Utah sample, we calculated mean read depth across all autosomal base pairs covered by >=12 reads in all members of the trio. We then assessed whether there was a correlation between mean read depth and the autosomal mutation rate in these samples. For each generation, we fit a linear model predicting read depth as a function of autosomal mutation rate, and do not find a significant association in either generation at a p-value threshold of 0.05 (second-generation p = 0.92, third-generation p = 0.073).

2) Another concern about paternal age effects is the extent to which outlier offspring may be driving the apparent rate variation across families. If the authors were to randomly sample half of the children from each family and run the analysis again, how much is the paternal age effect rank preserved? Alternatively, how much is the family rank ordering preserved if mutations are only called from a subset of the chromosomes?

This is an understandable concern. To address the issue of outlier samples impacting apparent variation in paternal age effects across families, we took the following approach. For each of the 40 CEPH/Utah families shown in Figure 3, we randomly sampled three-quarters of the family’s offspring. Given the family sizes of the CEPH/Utah cohort (median of 8 grandchildren) and manual inspection of the regressions for each family, we felt that this sampling strategy would remove the small numbers of possible outlier samples in each family without dramatically reducing the number of samples used for regression. We then fit the following regression on each subsampled family:

m = glm(autosomal_dnms ~ dad_age, family=poisson(link=”identity”))

Finally, we ranked each of the 40 subsampled families in order of increasing slope; as mentioned in the manuscript, the slope in each family represents the sum of both the paternal and maternal age effects. We repeated this procedure (random sampling followed by regression and re-ranking) 100 times, and aggregated the ranks for each family. In Figure 3—figure supplement 2, we have plotted the distribution of ranks (across 100 trials) for each of the 40 CEPH families. These distributions are ordered by the original ranks of the families, as determined using the full dataset and originally presented in Figure 3. We find that some families are indeed sensitive to possible outlier samples, as the ranks of some families are substantially changed after removing these outliers (Figure 3—figure supplement 2). For example, the distribution of ranks for family 26 appears to be approximately bimodal, suggesting that some families are quite sensitive to a small number of outliers. It is perhaps unsurprising that decreasing the number of data points in each family would change the ranks of certain families, as each family’s regression might become less precise, and potentially even more sensitive to small outliers.

Importantly, though, we note that for nearly all of the families, the median ranks after 100 simulations are very similar to the ranks inferred using the full, original dataset. Overall, these results suggest that our estimates of paternal age effect “ranks” for the 40 CEPH/Utah families are robust to possible outlier samples.

3) A factor regarding paternal age effects that is only mentioned briefly alluded to late in the paper are the differences in the intercept between families (Figure 3C). Do either the intercept or slope vary with the number of F2 children? Is there any (anti-)correlation between slope and intercept? It would seem odd if the intercept strongly impacts the slope since a low per-year rate probably should not correlate with a high initial rate at younger ages.

We do observe an anti-correlation between slopes and intercepts across the 40 CEPH/Utah families (Author response image 2). A significant negative correlation between slopes and intercepts is the expected consequence of fitting regression lines to data in which the mean of the independent variable is greater than 0. For example, we would expect to observe a negative correlation between slopes and intercepts if the third-generation DNM counts in all families were randomly scattered along the same y = a + bx line (where x represents paternal age at birth, and a and b are fixed constants), and regressions were fit for each family separately. Since a random distribution of DNM counts in each family would also produce a negative correlation between slopes and intercepts, it is possible that stochastic noise in the CEPH families’ DNM counts might be contributing to some of the variability in slopes we observe, as well as the resulting negative correlation with intercepts. Indeed, the confidence intervals surrounding slope point estimates in CEPH families are occasionally quite wide (Figure 3D), demonstrating the uncertainty in some of these estimates.

**Author response image 2. respfig2:** Anti-correlation between slope and intercept. For each CEPH/Utah family, we fit a linear model predicting DNM counts as a function of paternal age (see Figure 3). We then assessed whether the slopes and intercepts of these regressions were correlated; overall, slope and intercept point estimates are negatively correlated in CEPH/Utah families (p < 2.2e-16).

However, this noise alone is unlikely to produce the significant inter-family variability we find in our observed third-generation DNM counts, and we feel confident that our results represent true biological differences between families. Specifically, we can randomly distribute DNM counts in all third-generation CEPH/Utah individuals by drawing a single value from a Poisson distribution – with λ = a + (b * x), where x represents paternal age at birth, a = 15, and b = 1.72 – for each third-generation sample. The values of a and b were chosen to match the intercept and slope point estimates of the regression predicting autosomal DNM counts as a function of paternal age, using the full set of DNMs in the third generation.

If we test for inter-family variability in these simulated data, we find that a “family-aware” model is *not* a significantly better fit to the data than a “family-agnostic” model. We have included a Jupyter notebook (Reviewer Response Notebook, filename “response_figures.ipynb”) that includes code needed to recreate the figures and analyses presented in the main reviewer response, available at the following GitHub site: https://github.com/quinlan-lab/ceph-dnm-manuscript (copied archived at https://github.com/elifesciences-publications/ceph-dnm-manuscript/tree/master/notebooks). Additionally, we have added a caveat about the large confidence intervals in some families’ slope estimates, as well as the possible contribution of stochastic noise to these estimates, in the subsection “Identifying gonadal, post-primordial germ cell specification (PGCS) mosaicism in the second generation”.

To the reviewers’ other points, we do not observe any correlation between the number of siblings in a particular family (i.e. the number of third-generation individuals) and either the slope or intercept measured in that family (Author response image 3).

**Author response image 3. respfig3:** Lack of correlation between sibling number and either slope or intercept. For each CEPH/Utah family, we fit a linear model predicting DNM counts as a function of paternal age (see Figure 3). We then assessed whether the number of third-generation siblings in these families was predictive of either the (**a**) slope or (**b**) intercept point estimate in the regression. Neither slope (p = 0.654) or intercept (p = 0.718) are significantly associated with sibling number.

4) Another analysis that is informative about the cellular stage at which mutations occurred is to examine, for mutations found in F1, what fraction of these are transmitted to F2s. Putative DNMs could in fact be present in blood but not in the germline, and the study design here makes it easy to identify the fraction of such DNMs. A complexity of this is the use of multi-sample genotype calling. Perhaps dividing the genotype calling into a set called using the P0 and F1 generations only and comparing the resulting DNMs to those found in F2s (with calling in everyone) would ensure that the set of DNMs aren't biased towards those also present in the germline of F1s.

We agree that the untransmitted DNMs are an interesting class of potential mutations, and could represent somatic DNMs in the second generation. Thus, we returned to our original de novo mutation calls and counted the number of DNMs observed in each second-generation individual that were *not* transmitted to the third generation. For this analysis, we did not consider any second-generation individuals without sequenced children in the CEPH/cohort. Using a filtering strategy similar to the one described in the Materials and methods section (no likely or possible carriers, GQ >= 20 in the second-generation individual and both parents, DP >= 12 in the second-generation individual and both parents), we observed 3,919 untransmitted DNMs.

The counts of filtered untransmitted DNMs were not normally distributed across second-generation individuals. The median number of untransmitted DNMs per second-generation sample was 30, but four samples had substantially elevated counts of untransmitted DNMs (180, 187, 223, and 1,098 DNMs). For the purposes of this analysis, we removed these samples from further consideration, leaving a total of 2,231 untransmitted DNMs. The distribution of allele balances (fraction of reads supporting the alternate, de novo allele) in the filtered untransmitted DNMs (median AB = 0.182) was quite different than in the DNMs transmitted at “high-quality” to the F2 generation (median AB = 0.487, Author response image 4).

**Author response image 4. respfig4:** Allele balance distributions in transmitted and untransmitted DNMs. Allele balance was calculated as the fraction of reads supporting the alternate (i.e., de novo) allele at a particular site. As there are substantially more transmitted than untransmitted DNMs in the plot, the y-axis is shown as the normalized count of DNMs.

This substantial difference in allele balance could reflect the fact that the untransmitted DNMs are, by and large, false positives. However, it is also possible that the untransmitted DNMs are post-zygotic mutations that occurred following the fertilization of the F1’s embryo, present exclusively in somatic cells and absent from the germline. To discriminate false-positive from possible post-zygotic untransmitted DNMs, we visually examined a subset of the 2,231 untransmitted DNMs using the Integrative Genomics Viewer (IGV). Following visual inspection of 200 randomly sampled untransmitted DNMs, we found 130 likely false positive DNMs, likely a result of mapping artifacts, genotyping error, and other possible factors. Since the majority of untransmitted DNMs appear to be false positives, it is difficult to estimate the true fraction of each sample’s DNMs that are post-zygotic. However, this result suggests that post-zygotic (somatic) DNMs do exist within the set of untransmitted de novo mutations identified in the second-generation. Given the scope of this paper, we have elected to save a more detailed treatment of possible untransmitted post-zygotic mutations for a future analysis, though it presents an interesting direction for future work.

The reviewers also raise an important point regarding our differential power to detect DNMs in the second and third generations due to the multi-generational structure of the pedigrees. Because transmitted DNMs are (by their nature) present in more than one sample, a variant caller can integrate these multiple observations of the mutation into its posterior probability that the mutation is “real.” However, given the number of samples in the CEPH dataset and the complexities/time involved in re-running the full variant calling pipeline on two distinct sets of samples, we chose not to perform additional rounds of genotype calling. We anticipate that using the CEPH/Utah sequencing data, future analyses could address this important question.

5) In the additional data files, I could not find the age of all the individuals. It would be informative if the age information can be provided on the pedigree diagrams or as a separate files with identifiers.

Currently, all of the second- and third-generation individuals have associated paternal and maternal ages at birth in the ‘second_gen.dnms.summary.csv’ and ‘third_gen.dnms.summary.csv’ files, respectively. However, our IRB precludes us from providing ages and/or exact birth dates for every sample in the dataset, as this is more sensitive, identifiable information.

6) Will the sequencing data generated here be posted to the SRA or dbGaP?

The sequencing data for all 603 CEPH/Utah individuals (as well as a joint-called VCF) will be uploaded via dbGaP under controlled access. We have begun depositing these data in the Sequence Read Archive and dbGaP, though we don’t yet have an accession number for our data submission.

7) The authors state that "a gamete sampled from a younger father is more likely to possess a DNM that will recur in a future child." This doesn't seem correct as stated – every gamete should be equally likely to possess a DNM that will recur in a future child, independent of parental age. I believe what is meant is that a particular DNM sampled from the child of a young parent is more likely to be shared with a sibling than a DNM sampled from the child of an older parent.

The reviewer is correct; our original wording of this phrase is not accurate as stated. We have updated the subsection “Identifying gonosomal mosaicism in the second generation”, to reflect the reviewer’s correction.

Other questions and suggestions:1) One concern is in defining a site as "callable", which is of course not strictly binary. It would be good to take sequencing depth into consideration when deciding callability. Ideally this would also factor into both the FPR and MHR values. Given the three-generation study design, there is a greater opportunity to perform these analyses in a more detailed manner than in trio studies and thus to better estimate/model these rates.

We agree that “callability” is not a binary quality, and have attempted to address concerns about variable sequencing depth by investigating the correlation between mutation rates and average sequencing depth in CEPH/Utah samples (see response to Major revision #1).

One additional informative experiment might be to compare the MHR in CEPH/Utah families using either the 30X or 60X data in the 8 first-generation grandparents who were re-sequenced at higher depth. We hypothesize that using the 60X data, we might identify even more instances of “missed heterozygotes,” in which a grandparent is heterozygous for a particular variant that goes undetected in the second-generation, only to appear as an ostensibly de novo mutation in the third generation. However, this experiment would require re-running the full variant calling pipeline using the 60X data for these first-generation samples, and given the scope of this manuscript, we feel that it is best left for a future analysis.

2) Another factor to analyze is the use of multi-sample genotype calling and its potential to bias against the identification of non-mosaic (singleton) DNMs. Perhaps the 60x vs. 30x analysis can help estimate the rates of missing singletons in a way that is distinct from the MHR analysis.

The reviewers again raise an interesting hypothesis, which has implications for other large, family-based sequencing studies: are singleton (i.e., untransmitted) DNMs less likely to be identified in a joint-genotyping approach, as there is inherently less evidence for those DNMs in the rest of the cohort? Given the scope of this paper, we have not addressed this concern explicitly. However, we expect that in the future, researchers could make use of the CEPH/Utah dataset to address this more robustly.

Overall, we do not believe that the 60X and 30X data would be particularly useful for estimating a rate of “missing” singleton DNMs. The approach suggested in reviewer comment #3 (separating the CEPH/Utah families into groups of first/second and second/third-generation samples, following by joint-genotyping of each group separately) might be more fruitful for this analysis. Instead, the 60X and 30X data allow us to carefully estimate the fraction of apparently “real” singleton DNMs that are, in fact, likely inherited mutations that went undetected in a parent (see section entitled “Estimating a false positive rate for our de novo mutation detection strategy” in the Materials and methods section of the manuscript).

We hypothesize that if we had access to higher-depth (60X) sequencing data for *children* in the CEPH/Utah cohort, rather than grandparents, we might be able to use those deep sequencing data to better estimate the rates of missing singletons. Increased depth in the CEPH/Utah children could result in the increased sampling of alternate alleles; using the 30X data alone, it’s possible that these alleles may have been unobserved. Increased sampling of these alternate alleles could increase the sensitivity of the variant calling software, lead us to identify a larger number of true singleton DNMs, and obtain a better estimate of the fraction of singleton DNMs that go “missing” using only 30X data.

3) What is the range of the MHR? Is there significant variation with respect to MHR among families (in cases where P0 were genotyped)? Moreover, is there any enrichment or biases in mutational spectra seen using the MHR?

We have calculated the range of missed heterozygote rates across the CEPH families. Overall, the MHR is relatively low and consistent across these families (Author response image 5). Though the MHR is very low overall (~0.4%), we compared the mutation spectrum in the third-generation DNMs with grandparental evidence (i.e., DNMs that were removed as “missed heterozygotes”) to the filtered, high-quality third-generation germline DNMs, and did not find significant differences for any particular mutation types (Author response image 5).

**Author response image 5. respfig5:** Range of missed heterozygote rates across CEPH families. (**a**) For each unique set of second-generation parents and third-generation children, we counted the total number of DNMs in the third generation for which we saw evidence in the first generation (i.e., grandparents). The missed heterozygote rate (MHR) therefore represents the fraction of DNMs in each family that were likely “missed” in the second generation, as a percentage of the total number of DNMs identified in the third-generation children. (**b**) Comparison of mutation spectra in autosomal filtered germline third-generation DNMs (n=22,644) and autosomal third-generation DNMs that were removed due to evidence in a genotyped grandparent (n=83). No significant differences for particular mutation types were found at a false-discovery rate of 0.05 (Benjamini-Hochberg procedure) using a Chi-squared test of independence.

4) The authors mentioned that they have removed DNMs with likely "DNM carriers" in the cohort. Does this remove DNMs where the alternate alleles are observed in only the unrelated individuals or does it also include the related individuals?

We defined “carriers” as unrelated individuals who possess the DNM allele, and did not include individuals in the same immediate family as the sample with the DNM (i.e., siblings, parents, or grandparents) in our carrier observations. We have made this definition clearer in the Materials and methods (subsection “Identifying DNM Candidates”).

5) Other things to explore further related to parental age effects are: how do the conclusions change and/or can you detect similar variability in maternal age when analyzing phased DNMs? This may be underpowered, but for those families that share grandparents, if two brothers are in the F1 generation, do their paternal age effects differ?

As mentioned in the manuscript, given the low phasing rate in the third generation, we assessed inter-family variability using only the total autosomal counts of DNMs in each third-generation individual. However, we also attempted to identify similar variability using only the phased paternal or maternal de novo mutations. Using only the paternal de novo mutations, we performed a regression and ANOVA as follows:

m = glm(dad_dnms_auto ~ dad_age * family_id, family=poisson(link=”identity”))

anova(m, test=”Chisq”)

We find that the additive family_id term is significant in the model at a p-value threshold of 0.05 (p = 2.82e-4), though the interaction between dad_age and family_id is not (p = 0.402).

We also performed the regression using only the maternal DNMs:

m = glm(mom_dnms_auto ~ mom_age * family_id, family=poisson(link=”identity”))

anova(m, test=”Chisq”)

In many families, the number of children with 0 maternally phased DNMs renders the Poisson regression model unable to calculate coefficients. Therefore, we added a pseudo-count of 1 to all of the F2 individuals’ maternal DNM counts and re-ran the regression. Neither the additive family_id term (p = 0.221) nor the interaction between mom_age and family_id (p = 0.623) were significant at a p-value threshold of 0.05, likely due to the small numbers of phased maternal DNMs in each sample (range = 0-12).

The reviewers are correct that in this study, there are not many instances in which two brothers each had children whose DNA was sequenced. However, family ID 24 and family ID 19 (Supplementary file 1) each present an opportunity to investigate the paternal age effects of 2 brothers. Samples 426 and 444 are both members of family 24; in Figure 3, these two brothers and their children form the unique families “24_C” and “24_D.” To identify possible differences in paternal age effects between these brothers, we can fit a generalized linear model to a subset of the third-generation DNM counts that only includes families “24_C” and “24_D,” and run an ANOVA as done previously:

m = glm(autosomal_dnms ~ dad_age * family_id, family=poisson(link=”identity”))

anova(m, test=”Chisq”)

Following this test, we don’t find that a “family-aware” model is a better fit than a “family-agnostic” model at a p-value threshold of 0.05 (ANOVA p = 0.137), though this could be due in part to the small number of data points in each family, and the uncertainty surrounding their slope and intercept point estimates (Figure 3D). Indeed, in Figure 3D we can see that family “24_C” has the lowest slope point estimates of all 40 families, and the slope point estimate in family “24_D” is nearly identical to the median slope across all families.

We performed the same statistical test using a subset of the third-generation DNM counts that included only families “19_A” and “19_B.” These two families also contain a pair of brothers, who each had sequenced third-generation children in the CEPH dataset. Once again, a “family-aware” model is not a significantly better fit than a “family-agnostic” model (ANOVA p = 0.614), suggesting that there aren’t substantial differences in paternal age effects between these two brothers. This observation is supported by visually examining the point estimates for families “19_A” and “19_B” in Figure 3D, which appear to be quite similar.

Overall, however, it is difficult to confidently determine whether the above sets of brothers differ in their paternal age effects, given the small number of data points in each comparison. Indeed, there are many pairs of unrelated families that also do not appear to differ substantially in their paternal age effects.

6) For the parental age effect model the authors have correctly included "family-id", but for the rest of their analysis they have defined a "family" as the unique group of two F1 parents and their F2 offspring (e.g., Figure 3—figure supplement 1). Can the authors comment whether this might introduce biases in their analysis and filtering strategies, as some families are more related to each other than the rest?

For our analyses of inter-family variability, we defined “families” as the unique groups of second-generation parents and their third-generation offspring. Thus, in our regression models, the family_id term represents these 40 unique family IDs.

We note that in Figure 3D, there doesn’t appear to be any clear bias in terms of related families having very similar slopes, though we may be underpowered to detect differences between related second-generation individuals (see our response to comment #5 for an analysis of paternal age effects in two sets of brothers).

Additionally, we do not believe that the interconnected structure of some CEPH/Utah families would impact our filtering strategies. Our filters on depth, genotype quality, and allele balance, for example, should not be biased by possible relatedness between families.

Of course, if there truly are genetic modifiers of the mutation rate segregating in human populations, it is possible that more related families would have more similar parental age effects on DNM counts. In our manuscript, however, we are likely underpowered to detect such similarities, given the relatively small number of data points in each family.

7) Figure 4 shows that the number of DNMs shared with siblings does not appear to correlate with paternal age. Although it is seems unlikely to affect the result, it seems odd to report these as raw counts without correcting for the number of siblings the child has. It would be good to report the strength of the correlation between family size and shared DNM count and correct the shared counts for family size before testing for a correlation with paternal age.

We agree, and now report the lack of correlation between family size and shared DNM count (p=0.426, subsection “Assessing age effects on post-PGCS DNMs”). Given this lack of correlation, and the fact that all but one CEPH/Utah family has at least 8 children, we do not anticipate that sibling number would impact the correlation between shared germline mosaic DNM number and parental age at birth.

8) Why, from Figure 4B, are the differences in mutational spectra found in the post-PGCS mosaic analysis only based on 289/721 of these DNMs (presumably the phased ones)?

When searching for shared germline mosaic variants in the third generation, we identify all of the apparent DNMs in the third generation that are shared with at least one sibling. Thus, if we identify a particular de novo mutation that is shared by 3 siblings, that DNM would be represented 3 times in the set of 721 (720 in the updated version of the manuscript) post-PGCS DNMs. For our comparisons of mutation spectra, we did not want to count the same mutation multiple times if it was shared by siblings; the “289” number therefore reflects the number of *unique* autosomal sites (i.e. unique autosomal mutational events) in the list of 721 shared mosaic DNMs.

9) A minor but important consideration here is as a term, "post-PGCS", seems to include any mutations that arise following the establishment of the germ cells, but what seems to be the intended meaning is those mutations that arise during germ cell proliferation (or related). Rewording would aid understanding here.

We agree that the term “post-PGCS” is a bit imprecise, since “normal” single-gamete germline DNMs technically occur post-PGCS, as well. Therefore, we instead refer to the “post-PGCS” variants as “shared/germline mosaic DNMs” throughout.

10) For the mutational spectra analysis of gonosomal mosaic DNMs, this and other similar analyses consider each allelic class independently. Would power increase by analyzing the data as a whole using, say, a Chi-squared six degree of freedom test?

The reviewers are correct that a Chi-squared test with six degrees of freedom might offer greater power to detect significant differences in mutation spectra. However, for the purposes of our analyses, we were interested in identifying specific differences for each mutation class separately.

11) The authors have applied the same filters as DNMs for identifying post-PGCS mosaic variants. They seem to have filtered candidates based on VAF > 0.2. Might this filter may be too stringent for identifying the post-PGCS events?

We may be misunderstanding the reviewer’s concern here, but the post-PGCS variants should appear to be “normal” germline DNMs in third-generation individuals, but happen to be shared with other third-generation siblings. In other words, we would expect a post-PGCS DNM to be a heterozygous mutation present in every cell of the third-generation child; this is because the mutation actually occurred in a progenitor of the sperm or egg cell that ultimately fertilized the third-generation child’s embryo. To identify the post-PGCS mosaic variants, we search through all of the DNMs seen in the third generation, and find the DNMs that are shared by siblings. As a result, the same filters applied to the “normal,” single-gamete germline DNMs in the third generation (VAF >= 0.3) are applied to the post-PGCS variants.

12) To identify gonosomal mutations the authors have applied hard VAF cut off < 0.2, considering the number of cell divisions before PGC, would this threshold be a bit too low? Would their observation change significantly if they change the threshold to <0.3?

The reviewers raise an important concern here; namely, that if a post-zygotic mutation occurs very soon after fertilization of the embryo, it could be present in a large fraction of somatic cells, and manifest with VAF > 0.2. In the process of addressing this concern, we substantially improved our strategy for identifying gonosomal post-zygotic DNMs, and now use a phasing-by-transmission strategy rather than a VAF cutoff (see “Note from authors regarding post-zygotic mosaicism”). As a result, we no longer apply strict VAF cutoffs to the gonosomal mutations, and can identify gonosomal mutations present at VAF > 0.2.

13) What is the VAF distribution of candidate gonosomal mutations in F2? One of the filters they have used have VAF >=0.3 in F2. Might this threshold be too lax? For the gonosomal mutations that occur in F1, an expectation of a higher VAF of almost 0.5 in the F2 set seems reasonable.

The reviewers are correct that the candidate gonosomal mutations (which occurred during the post-zygotic development of the F1 individuals) should be present at high VAF (~0.5) in the F2 individuals, since these mutations should have been inherited as “normal” heterozygous mutations by the F2 children. We applied a VAF filter of >= 0.3, simply because we expect the VAF for true heterozygous variants to be approximately normally distributed about a mean of 0.5, with the VAF for most of these variants falling between 0.3 and 0.7.

14) In mosaic post-PGCS analysis: the authors have identified 32 events with supporting alleles in F1. Among these 32 mutations, do any occur in families were F1s are related? For example, do F1 19_A mom and 19_B dad share some of these mosaic mutations? If so is there any correlation in mutational burden in F1 with the age of P0?

Some of the post-PGCS mutations with supporting evidence in a parent did occur in families where second-generation individuals were related (including family IDs 19 and 24). However, none of these mutations were found in multiple second-generation individuals (i.e., they each occurred at a unique site).